# The Dissipative Photochemical Origin of Life: UVC Abiogenesis of Adenine

**DOI:** 10.3390/e23020217

**Published:** 2021-02-10

**Authors:** Karo Michaelian

**Affiliations:** Department of Nuclear Physics and Applications of Radiation, Instituto de Física, Universidad Nacional Autónoma de México, Circuito Interior de la Investigación Científica, Cuidad Universitaria, Mexico City, C.P. 04510, Mexico; karo@fisica.unam.mx

**Keywords:** origin of life, dissipative structuring, prebiotic chemistry, abiogenesis, adenine, organic molecules, non-equilibrium thermodynamics, photochemical reactions, 92-10, 92C05, 92C15, 92C40, 92C45, 80Axx, 82Cxx

## Abstract

The non-equilibrium thermodynamics and the photochemical reaction mechanisms are described which may have been involved in the dissipative structuring, proliferation and complexation of the fundamental molecules of life from simpler and more common precursors under the UVC photon flux prevalent at the Earth’s surface at the origin of life. Dissipative structuring of the fundamental molecules is evidenced by their strong and broad wavelength absorption bands in the UVC and rapid radiationless deexcitation. Proliferation arises from the auto- and cross-catalytic nature of the intermediate products. Inherent non-linearity gives rise to numerous stationary states permitting the system to evolve, on amplification of a fluctuation, towards concentration profiles providing generally greater photon dissipation through a thermodynamic selection of dissipative efficacy. An example is given of photochemical dissipative abiogenesis of adenine from the precursor HCN in water solvent within a fatty acid vesicle floating on a hot ocean surface and driven far from equilibrium by the incident UVC light. The kinetic equations for the photochemical reactions with diffusion are resolved under different environmental conditions and the results analyzed within the framework of non-linear Classical Irreversible Thermodynamic theory.

## 1. Introduction

There exist many proposals for the abiogenesis of the fundamental molecules of life (those found in all three domains) supported by a large body of empirical data for the exogenous delivery (comets, meteorites, and space dust) [1,2] or endogenous synthesis (atmospheric, ocean surface, warm ponds, hydrothermal vents) [3,4,5]. Free energy sources proposed for affecting synthesis include meteoric shock impact, electric discharge, temperature gradient, pH gradient, particle radiations, gamma rays, UV light, organocatalysis, micro forces, etc. [6,7,8]. A robust explanation of the origin of life, however, requires a clear understanding of not only how biologically important molecules spontaneously emerged, but also how they proliferated and evolved together into ever more complex dissipative structures, eventually leading to the global dissipative processes known as the *biosphere*, incorporating both biotic and abiotic components.

Proposals for the origin of life have considered a fortuitous emergence of a self-replicating or autocatalytic molecular system undergoing incipient Darwinian-type evolution based on selection of molecular stability, fidelity, or chemical sequestration. However, no detailed theory based on the Darwinian principle has yet been proposed and certainly no indulgent chemical reaction sets have yet been found.

Non-equilibrium thermodynamic theory, in particular Classical Irreversible Thermodynamics (CIT), developed by Théophile de Donder, Lars Onsager, Ilya Prigogine, Paul Glansdorff, Grégoire Nicolis, Agnessa Babloyantz, and others from the “Brussels school” has proven to be a very useful formalism for understanding living systems and their dynamics, including; the origin of life [9,10,11,12,13,14,15,16,17,18], the cell [19], cell differentiation [20], ecosystems [21,22], the biosphere [19,23,24,25,26] and even the synthesis of organic molecules detected in space [27].

In this paper, I employ CIT theory to analyze the abiogenesis of adenine from HCN in a water solvent environment under the imposed UVC photon flux prevalent at the origin of life in the Archean. The model consists of a set of non-linear photochemical and chemical reactions with diffusion occurring within a fatty acid vesicle, driven far from equilibrium by the impressed UVC light. If intermediate product molecules on route to the synthesis become catalysts for the chemical or photochemical reactions, then this leads to their proliferation, as well as to that of their final product. Because the system is non-linear, it has multiple stationary states of different product concentration profiles. The system evolves through these states in response to external fluctuation near a bifurcation. Selection is towards greater photon dissipative efficacy and is probabilistic; determined by fluctuations near instabilities and the widths of phase-space paths to conical intersections (Section 2.2) which lead to the intermediate molecules (Section 3). This evolution is subject to the *universal evolutionary criterion* of Glansdorff and Prigogine (Appendix A). This, along with auto- and cross-catalytic proliferation, provides a mechanism for evolution which may be termed *dissipative selection*, or more generally, *thermodynamic selection*. Dissipative structuring, dissipative proliferation, and dissipative selection, are the necessary and sufficient elements for a non-equilibrium thermodynamic framework from within which the origin and evolution of life can be explained in purely physical and chemical terms [10,11,13].

In Section 2 I briefly describe the photochemical transformation mechanisms important in the UVC dissipative structuring of molecules. In Section 3 I describe, and provide empirical evidence for, the framework employed here known as the Thermodynamic Dissipation Theory of the Origin of Life [10,11] which was the first to describe the origin of life as the dissipative structuring of the fundamental molecules and their complexes under the Archean UVC light. In Section 4 I give an explicit example of the photochemical dissipative structuring, proliferation, and selection of molecules on route to adenine. Finally, in Appendix A I provide the mathematical formalism of the non-equilibrium thermodynamics of Prigogine and co-workers needed to understand the dissipative structuring, proliferation, and evolution of life and how this would lead to increases in global solar photon dissipation, roughly corresponding to increases in entropy production.

## 2. Photochemistry

### 2.1. Quantum Selection Rules

Absorption by an organic molecule of a visible or UV photon of the required energy E=hν leads to an electronic spin singlet or triplet excited state. The width of the allowed transition ΔE is determined by the natural line width dependent on the natural lifetime Δt of the excited state, as given by the Heisenberg uncertainty relation ΔEΔt≥ħ. In condensed material or at high pressure, further broadening occurs due to deexcitation through collisions with neighboring molecules, reducing further the lifetime. There is also a broadening due to the Doppler effect which increases with temperature. Contributing most to the broadening for some organic molecules, however, is the coupling of electronic degrees to the vibrational degrees of freedom of the molecule (vibronic or nonadiabatic coupling) through an excited state and ground state potential energy surface degeneracy known as a *conical intersection*.

Excitation to the triplet state is a spin forbidden transition but can occur due to spin-orbit coupling or interaction with a paramagnetic solvent molecule, for example oxygen in its spin-triplet ground state. Under laboratory conditions, and for organic molecules, however, the singlet state is favored over the triplet state by ∼1000:1. Moreover, since electronic excitations are affected by the electronic dipole transition which is first order in the coordinates *x* (i.e., the dipole moment is an odd function f(x)≠f(−x)), and since an additional quantum selection rule is that transitions must be symmetric, the symmetries of the wavefunctions of the molecule in the initial and final state must be different (e.g., even → odd) giving rise to the electronic angular momentum selection rule Δl=±1. For example, a 1S→2S transition is forbidden while a 1S→2P transition is allowed.

### 2.2. Conical Intersections

The Born-Oppenheimer approximation in molecular structure calculations assumes independence of the electronic and nuclear motions. However, such an approximation is obviously not valid for chemical reactions where nuclear reconfiguration is coupled to electronic redistribution and particularly not valid for photochemical reactions where the potential energy surface of an electronic excited state is reached.

Conical intersections are multi-dimensional seams in nuclear coordinate space (Figure 1) where the adiabatic potential energy surface of the electronic excited state becomes degenerate with the potential energy surface of the electronic ground state of the same spin multiplicity, resulting from a normally barrier-less out of plane distortion of the nuclear coordinates (e.g., bond length stretching, rotation about a bond [28], or frustrated H-atom ejection). A common distortion of the nuclear coordinates for the excited state of the nucleobases is ring puckering as shown for adenine in Figure 1. This multi-dimensional seam, defining the energy degeneracy, allows for rapid (sub-picosecond) radiationless deexcitation of the photon-induced electronic excited state, distributing the electronic energy over nuclear vibrational modes of the molecule as either a prelude to a photochemical transformation, or to the rapid dissipation of the energy into the solvent (internal conversion) leaving the molecule in its original ground state ready for another photon absorption event.

The conical intersection seams thus define the photoisomerization or photoreaction products that can be reached after an electronic excitation event. Since conical intersections are located energetically down-hill from the Franck-Condon region, the direction and velocities of approach of the nuclear coordinates to a conical intersection are important in defining the outcome [28]. For example, it is known that for the molecule retinal in rhodopsin, the photoexcited molecule reaches the conical intersection extremely fast (75 femtoseconds) implying that the conical intersection must be peaked (inverted cone-like on the excited state potential energy surface) and, overwhelmingly, only one reaction product is reached, which for the case of retinal, as well as for the fundamental molecules of life, is the original ground state configuration [33]. A more extended seam with different minima can lead to different reaction products [34] such as those intermediates on route to the photochemical synthesis of adenine which will be described in Section 4. The final products of the photochemical synthesis (the fundamental molecules of life), however, often have a *peaked* conical intersection to internal conversion so that they become the *final* and *photo-stable* product of dissipative structuring in the relevant region of the solar spectrum.

It has been a recurrent theme in the literature that the rapid (sub-picosecond) deexcitation of the excited nucleobases due to their conical intersections had evolutionary utility in providing stability under the high flux of UV photons that penetrated the Archean atmosphere [35,36] since a peaked conical intersection reduces the lifetime of the excited state to the point where further chemical reactions are improbable. However, photo-stability is never complete, and photochemical reactions under UVC light do still occur for the fundamental molecules of life, particularly after excitation to the long-lived triplet state, for example in the formation of cyclobutane pyrimidine dimers in RNA and DNA. An apparently more optimal and simpler solution for avoiding radiation damage with its concomitant degradation in biological function, therefore, would have been the synthesis of molecules transparent to, or reflective to, the offending UV light. From the perspective of the thermodynamic dissipation theory for the origin of life employed here, however, a large antenna for maximum UVC photon absorption and a peaked conical intersection for its rapid dissipation into heat are, in fact, precisely the design goals of dissipative structuring.

### 2.3. Excited State Reaction Mechanisms

The photochemistry of molecules in electronic excited states is much richer than the thermal chemistry of their ground state, because; (1) the absorbed photon energy allows very endothermic reactions to occur, (2) anti-bonding orbitals can be reached, allowing reactions to occur which are prohibited in the ground state, (3) triplet states can be reached from the electronic excited state, allowing intermediates that cannot be accessed in thermal reactions, (4) electronically excited molecules are often converted into radicals, making them much more reactive. For example, a molecule in its excited state can be a much stronger oxidizer or reductor with a pK${}_{a}$ value substantially different from that of its ground state (e.g., if the pK${}_{a}$ value becomes more acidic, proton transfer to an acceptor solvent water OH${}^{-}$ ion becomes much more probable). Singlet excited states have a particularly rich chemistry, while triplet states have a more restricted chemistry. This richness in photochemistry is, in itself, yet another strong argument in favor of the suggestion that the complex molecules of life arose out of photon-induced reactions occurring at the surface of the ocean rather than out of thermal reactions occurring at the bottom of the ocean.

Photochemical processes that arise after photon-induced excitation can be classified into disassociations, rearrangement, additions and substitutions. Specific processes within these classes constitute particular mechanisms for molecular transformation which could have been employed in the photochemical dissipative structuring of the fundamental molecules at the origin of life under the Archean UVC photon flux. Indeed, these mechanisms still occur today in many important photochemical processes of life, albeit in the near UV or visible regions of the spectrum and through more complex biosynthetic pathways.

The photochemical transformations listed above generally have a strong dependence on wavelength due to the particular absorption characteristics of the inherent chromophores of the precursor molecules. However, it is not only the absorption coefficient of the chromophore which is important since within a given wavelength region there may be two or more such molecular transformational processes in competition, and therefore the particular conformation of the electronic ground state before excitation may be relevant. This conformation could depend on the temperature, viscosity, polarity, ionic strength and pH of the solvent, all of which are determinant in the yields of the final photoproducts.

Some of the molecular transformations mentioned above do not belong exclusively to the domain of photochemical reactions but can also occur through thermal reactions at high temperature, albeit with lower yield and less variety of product. Therefore, as empirically well established, some of the fundamental molecules of life could have been produced through thermal mechanisms without recourse to the incident light, for example at ocean floor hydrothermal vents. However, as emphasized in the Introduction, the mere efficient synthesis of the fundamental molecules is not sufficient to bootstrap the irreversible dissipative process known as life. The continuous dissipation of an external thermodynamic potential is a necessary condition for the structuring, proliferation, and complexation of life, as it is for any sustained irreversible process, and this is discussed at length in the following section.

## 3. The Thermodynamic Dissipation Theory of the Origin of Life

There are only two classes of structures in nature: *equilibrium structures* and *non-equilibrium* structures. Equilibrium structures arise naturally and their synthesis from arbitrary distributions of material can be described through the minimization of an internal thermodynamic potential, for example, a crystal structure, or a lipid vesicle, arising from the minimization of the Gibbs potential at constant temperature and pressure. The second class is that of non-equilibrium structures (or processes) known as *dissipative structures* which also arise naturally, but through the optimization of the dissipation of an externally imposed generalized thermodynamic potential [37], for example the “spontaneous” emergence of convection cells arising to increase the thermal dissipation at a critical value of an externally imposed temperature gradient, or the water cycle which arises to dissipate the incident solar photon spectrum.

Life, although incorporating equilibrium structures, is fundamentally a non-equilibrium process and therefore its origin, proliferation and evolution are wholly determined by the dissipation of one or more thermodynamic potentials from its environment. Boltzmann recognized this almost 125 years ago [38,39] and suggested that life dissipates the solar photon potential. Present-day life dissipates other thermodynamic potentials accessible on Earth’s surface, for example, chemical potentials available in organic or inorganic molecules or available in concentration, temperature, or pH gradients at hydrothermal vents. However, all ecosystems in which such organisms are embedded ultimately depend, or depended, on the dissipation of the solar photon spectrum.

At the origin of life around 3.9 thousand million years ago, UV photons provided approximately three orders of magnitude more free energy for dissipation as compared to that available from volcanic activity (hydrothermal vents), electric discharge, or meteoritic impact, combined [6,7,8], irrespective of a more radioactive Archean Earth. This UV solar photon flux was continually available at Earth’s surface during the Archean and thus could have provided the dissipative potential for not only molecular synthesis of the fundamental molecules, but also for molecular proliferation and the evolution towards complex structures of increasingly greater photon dissipation.

We have identified the long wavelength part of the UVC region (∼205–285 nm), plus the long wavelength part of the UVB and short wavelength part of the UVA regions (∼310–360 nm), as the thermodynamic potential which probably drove the dissipative structuring, proliferation, and evolution relevant to the origin of life [10,11]. This light prevailed at Earth’s surface from the Hadean, before the probable origin of life near the beginning of the Archean (∼3.9 Ga), and for at least 1000 million years [35,40,41] (Figure 2) until the formation of an ozone layer when natural oxygen sinks (for example, free hydrogen and Fe${}^{+2}$) became overwhelmed by organisms performing oxygenic photosynthesis. A strong argument for the relevance of this particular region of the solar spectrum, corresponding to the Archean atmospheric transparency at the origin of life, is that longer wavelength photons do not contain sufficient free energy to directly break double covalent bonds of carbon-based molecules, while shorter wavelengths contain enough energy to destroy these molecules through successive ionization or fragmentation.

Empirical evidence supports our conjecture of the dissipative structuring of the fundamental molecules of life under these wavelengths; first, the maximum in the strong absorption spectrum of many of these molecules coincides with the predicted window in the Archean atmosphere (Figure 2) [12]. Secondly, many of the fundamental molecules of life are endowed with *peaked conical intersections* (Section 2.2) giving them a broad absorption band and high quantum yield for rapid (picosecond) dissipation of the photon-induced electronic excitation energy into vibrational energy of molecular atomic coordinates, and finally into the surrounding water solvent [28]. Perhaps the most convincing evidence of all, however, is that many photochemical routes to the synthesis of nucleic acids [42], amino acids [43], fatty acids [17], sugars [44], and other pigments [12] from common precursor molecules have been identified at these wavelengths and the rate of photon dissipation within the Archean window generally increases after each incremental step on route to synthesis, a behavior strongly suggestive of dissipative structuring [14,17] (Appendix A).

Figure 3 describes why precursor molecules will gradually transform (evolve) towards structures of greater photon dissipative efficacy under the impressed UVC photon spectrum of the Archean. Even though the product molecule often (but not always) has a lower Gibb’s free energy than that of the precursor molecule from which it evolved, the evolution is not spontaneous if there are large energy barriers between configurations. Coupling of the reactions to the impressed UV photon potential (photochemical reactions) allows the transformation to proceed over the barriers at a rate dependent on photon intensities at the different wavelengths of maximum absorption I$\left(\lambda_{max}\right)$ for the two structures, and on the widths of the phase-space paths leading to the particular conical intersection on the electronic excited state potential energy surface (Section 2) promoting molecular transformation. Backward transformations, or transformations to other possible structures, under the UV light are less probable if the molecule has a smaller quantum efficiency (smaller conical intersection) to that transformation as compared to its quantum efficiency for internal conversion.

Furthermore, product molecules acting as auto- or cross-catalysts (or photosensitizers) for the reactions (or photochemical reactions) will naturally increase their representation at the expense of other products and thus come to dominate the population. These non-linearities, in fact, lead to multiple stationary states of different molecular concentration profiles for the chemical and photochemical reaction system. External fluctuations can lead the system from one stationary state to another, the stability of a given stationary state being similarly related to its overall photon dissipative capacity under the given incident spectrum (Figure 3). The final products (fundamental molecules of life) will therefore have the largest quantum efficiencies for internal conversion (dissipation) or catalytic proton transfer, consistent with the prevailing surface light spectrum. Macroscopic evolution (ensemble averaged), is therefore towards greater dissipative concentration profiles of the product molecules. It is therefore suggested that it was the efficacy of UV photon dissipation into heat that provided the driving force for the evolution through dissipative structuring of organic molecules and their macroscopic concentration profiles during the Archean. The nonlinear non-equilibrium thermodynamics of this selection process, which I call *dissipative selection*, or more generally *thermodynamic selection*, is described in detail in Appendix A.

Therefore, in contradistinction to the generally held view that UV wavelengths were detrimental to early life and thereby induced extreme selection pressure for mechanisms or behavioral traits that protected life from, or made life tolerable under, these photons [35,36,45,46], it is argued here that these wavelengths were not only fundamental to the photochemical synthesis of life’s first molecules (as suggested with increasing sophistication by Oparin [47], Haldane [48], Urey [49], Sagan [50] and Mulkidjanian [36] and supported experimentally by Baly [51], Miller [52], Oro and Kimball [53], Ponnamperuma et al. [54,55,56], Ferris and Orgel [42], and Sagan and Khare [43] as well as others) but that this UV light was fundamental to the origin and early existence of the entire thermodynamic dissipative process known as “life”, comprising of synthesis, proliferation, and evolution (Figure 3) leading to concomitant increases in biosphere photon dissipation over time.

Thus, rather than requiring refuge or protection from this UV light, it is argued here that UV-induced molecular transformations providing innovations which allowed early molecular life to maximize UV exposure; e.g., buoyancy at the ocean surface, larger molecular antennas for capturing this light, increases in the width of the wavelength absorption band, and peaked conical intersections to internal conversion providing extraordinarily low antenna dead-times, all would have been selected for [57] through dissipative selection based on non-equilibrium thermodynamic principles described above and detailed in Appendix A. There, in fact, exists empirical evidence suggesting selection for traits optimizing UV exposure for particular amino acids complexed with their RNA or DNA cognate codons or anticodons, particularly for those amino acids displaying the strongest stereochemical affinity to these [57]. This has led us to suggest [18,57] that UVC photon dissipation was the basis of the initial specificity in the nucleic acid—amino acid association during an early stereochemical era [58]. This provides new light on the origin of information translation from nucleic acid to amino acid, which is one of the enduring mysteries of molecular biology [59].

Finally, UVC photons provide orders of magnitude more free energy and many more pathways for carbon covalent bond transformation of precursor molecules than do thermal reactions (Section 2).

From the perspective of the *thermodynamic dissipation theory*, the origin of life was therefore not a scenario of organic material organization driven by natural selection leading to “better adapted” organisms, or to greater chemically stability (e.g., UV resistant organisms), but rather a scenario of the dissipative structuring of material under the impressed UV solar photon potential leading to a structuring of material in space and time (processes) in such a manner so as to provide a more efficient route to the dissipation of the externally impressed solar photon potential. Similar dissipative synthesis of an ever larger array of photochemical catalysts and cofactors, would allow ever more complex processes such as biosynthetic pathways to emerge through this thermodynamic dissipative selection to promote the synthesis of novel pigments for dissipating not only the fundamental UVC and other UV regions, but the entire short wavelength region of the solar photon spectrum [12,27], eventually reaching the red edge (∼700 nm), which is the approximate limit of biological photon dissipation on Earth today.

The thermodynamic dissipation theory for the origin of life [11,13] as summarized above, employed as the framework here, defines life as; *the dissipative structuring, proliferation, and evolution of molecular pigments and their support structures from common precursor carbon-based molecules under the impressed short wavelength solar photon potential for performing the explicit thermodynamic function of dissipating this light into long wavelength infrared light (heat)*. The external photon potential supplied continuously by the environment, and its dissipation into heat by the spontaneously assembled dissipative structures, are both integral components necessary for understanding life.

## 4. Example: The Dissipative Structuring of Adenine

### 4.1. The Model

HCN is a common molecule found throughout the cosmos and its production during the Hadean and Archean on Earth was probably a result of the solar Lyman alpha line (121.6 nm) photo-lysing N${}_{2}$ in the upper atmosphere which then attacks CH or CH${}_{2}$ to form HCN [60], or the UV (145 nm) photolysis of CH${}_{4}$ leading to a CH${}^{*}$ radical which attacks N${}_{2}$ [60]. HCN and its hydrolysis product formamide are now recognized as probable precursors of many of the fundamental molecules of life, including nucleic acids, amino acids, fatty acids [44], and even simple sugars [61,62]. As early as 1875 E. Pflüger suggested that life may have followed from “cyanogen compounds” [63]. The ubiquity of different chemical and photochemical routes from HCN to the fundamental molecules discovered over the last 60 years has led to the suggestion of an “HCN World” [64,65] occurring before the postulated “RNA World” [66].

The synthesis of adenine from HCN has been studied by numerous groups since the first experimental observations of the chemical reaction at high temperatures by Oró in 1960 [67] and photochemically at moderate temperatures by Ferris and Orgel in 1966 [42,68,69,70,71] (Figure 4). Adenine is a pentamer of HCN and the overall reaction from 5 HCN to adenine is exothermic (ΔG=−53.7 kcal mol${}^{-1}$ [70]) but presents several large kinetic barriers which can be overcome at high temperatures or at low temperatures if UV photons are absorbed (Figure 3). The reactions on route to adenine are in competition with hydrolysis and UV lysis, and these relative rates are dependent on concentrations, temperature, pH, metal ion- and product- catalysis, and the wavelength dependent intensity of the incident UV spectrum. The complexities involved in the photochemical reactions leading to adenine have been studied by Sanchez et al. [68,69].

**Figure 4 entropy-23-00217-f004:**
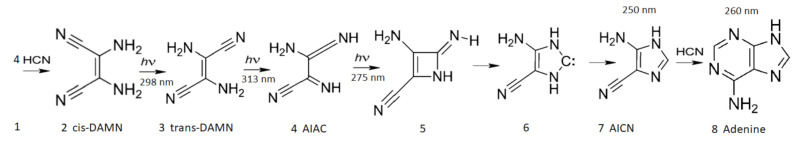
The photochemical synthesis of adenine from 5 molecules of hydrogen cyanide (HCN) in water, as discovered by Ferris and Orgel (1966) [42,71]. Four molecules of HCN are transformed into the smallest stable oligomer (tetramer) of HCN, known as cis-2,3-diaminomaleonitrile (cis-DAMN) (2), which, under a constant UVC photon flux isomerizes into trans-DAMN (3) (diaminofumaronitrile, DAFN) which may be further converted on absorbing two more UVC photons into an imidazole intermediate, 4-amino-1H-imidazole-5-carbonitrile (AICN) (7). Hot ground state thermal reactions with another HCN molecule or its hydrolysis product formamide (or ammonium formate) leads to the purine adenine (8). This is a microscopic dissipative structuring process (Figure 3) which ends in adenine [14], a pigment with a large molar extinction coefficient at 260 nm and a peaked conical intersection which promotes the dissipation of photons at the wavelength of maximum intensity of the Archean solar UVC spectrum (Figure 2). Adapted from Ferris and Orgel (1966) [42].

An apparent difficulty exists with respect to the synthesis of the purines from HCN in that, for dilute concentrations of HCN (<0.01 M), hydrolysis of HCN occurs at a rate greater than its polymerization, e.g., its tetramization (step 1 to 2, Figure 4), the first required step on route to adenine. Hydrolysis is proportional to the HCN concentration whereas tetramization is proportional to the square of the concentration [68]. Stribling and Miller [72] estimated that atmospheric production of HCN and subsequent loss to hydrolysis and recycling through thermal vents, would have led to ocean concentrations, at neutral pH, of no greater than about $1.0\times 10^{-12}$ M at 100 ${}^{\circ }$C and $1.0\times 10^{-4}$ M at 0 ${}^{\circ }$C for an ocean of 3 Km average depth. This led Sanchez, Miller, Ferris, and Orgel [68,73] to conclude that eutectic concentration of HCN (through freezing of the water solvent) would have been the only viable route to synthesis of the purines, and this is the primary reason subsequent analyses favored a cold scenario for the origin of life [74,75,76], not withstanding the geochemical evidence to the contrary, and even though this severely reduces all reaction rates and inhibits diffusion.

However, it is now known that the top ∼100 μm of the ocean surface (known as the microlayer) is a unique environment with organic material densities as large as 10${}^{4}$ times that of bulk water below. This is due to lowering of the free energy of fatty acids and other amphipathic molecules at the air/water interface, as well as Eddy currents and air bubbles from raindrops bringing organic material to the surface [77,78]. Furthermore, it has been shown that even though HCN is very soluble in water (and even in non-polar solvents), it tends to concentrate at a water surface and is observed to align itself through a dipole-dipole interaction in such a manner to facilitate polymerization. Molecular dynamic simulations of HCN in water have shown that it can form patches of significantly higher density in both the lateral and vertical dimensions at the surface, due to this strong inter-molecular dipole-dipole interaction [79].

Rather than invoking eutectic concentration at low temperature to increase the solute HCN concentration to values sufficient for significant adenine production, here we assume instead the existence at the surface of fatty acid vesicles of ∼100 μm diameter (results are relatively independent of diameter) which would allow the incident UVC light, as well as small molecules such as HCN and H${}_{2}$O, to enter or leave relatively unimpeded by permeating its bi-layer wall (Figure 5), while trapping within the vesicle the photochemical reaction products due to their larger sizes and larger dipole moments (Table 1). This would allow these molecules, as well as the heat from their UV photon dissipation, to accumulate within the vesicle.

The existence of amphipathic hydrocarbon chains, which through Gibb’s free energy minimization spontaneously form lipid vesicles at the ocean surface, is a common assumption in origin of life scenarios [80,81,82] and their abiotic production during the Archean could be attributed to heat activated Fischer–Tropsch polymerization of smaller hydrocarbon chains such as ethylene at very high temperatures at deep ocean hydrothermal vents, or, more likely, to dissipative structuring under UVC photons of HCN and CO${}_{2}$ saturated water at moderate temperatures on the ocean surface [17]. In order to maintain vesicle integrity at the hot surface temperatures considered here of ∼80 ${}^{\circ }$C these fatty acids would necessarily have been long (∼18 C atoms) and cross linked through UVC light which improves stability at high temperatures and over a wider range of pH values [17,83]. There is, in fact, a predominance of 16 and 18 carbon atom fatty acids in the whole available Precambrian fossil record [84,85].

In the following subsection I present a simplified out-of-equilibrium kinetic model for our 5HCN → adenine photochemical reaction system occurring within a fatty acid vesicle floating within the surface microlayer of a hot (∼80 ${}^{\circ }$C [86,87,88]) Archean ocean under the UV surface spectrum of Figure 2. It is assumed that that the system is under a diurnal 8 h flux of sunlight followed by an 8 h period of darkness during which thermal reactions occur but not photochemical reactions. The system is assumed to be perturbed by the existence of sparse patches of relatively high concentration (0.1 M) of HCN and formimidic acid (Fa) (a photon-induced tautomer of its hydrolysis product formamide (F)) into which our vesicle is assumed to drift for a brief period (120 s) only once during 30 Archean days.

### 4.2. The Kinetic Equations

Nomenclature, chemical formula, and abbreviations used throughout the text, for the concentrations of the participating chemical species of the photochemical reactions leading to adenine, as shown in Figure 4, along with their photon extinction coefficients, dipole moments and polar surface area, both related to their permeability of the vesicle walls, are given in Table 1.

**Table 1 entropy-23-00217-t001:** Nomenclature, chemical formula, abbreviation in the text and in kinetic equations, position in Figure 4, wavelength of maximum absorption $\lambda_{max}$ (within the spectrum of Figure 2), molar extinction coefficient at that wavelength $\epsilon_{max}$, electric dipole moment μ, and the topological polar surface area (TPSA), of the molecules involved in the photochemical synthesis of adenine. Values marked with “*” are estimates obtained by comparing to similar molecules since no data is available in the literature.

Name	Chemical Formula	Abbrev. in Text	Abbrev. in Kinetics	Figure 4	$\lambda_{\max}$ nm	$\epsilon_{\max}$ M${}^{-1}$ cm${}^{-1}$	μ [D]	TPSA [Å${}^{2}$]
hydrogen cyanide	HCN	HCN	H	1			2.98	23.8
formamide	H${}_{2}$N-CHO	formamide	F		220	60 [89,90]	4.27 [91]	43.1
formimidic acid	H(OH)C=NH	formimidic acid (trans)	Fa		220	60	1.14 [91]	43.1 *
ammonium formate	NH${}_{4}$HCO${}_{2}$	ammonium formate	Af				+/−, 2.0 *	41.1
diaminomaleonitrile	C${}_{4}$H${}_{4}$N${}_{4}$	cis-DAMN (DAMN)	C	2	298	14,000 [92]	6.80 [93]	99.6
diaminofumaronitrile	C${}_{4}$H${}_{4}$N${}_{4}$	trans-DAMN (DAFN)	T	3	313	8500 [92]	1.49 [93]	99.6
2-amino-3-iminoacrylimidoyl cyanide	C${}_{4}$H${}_{4}$N${}_{4}$	AIAC	J	4	275	9000 [68,71]	1.49	99.6 *
4-aminoimidazole-5-carbonitrile	C${}_{4}$H${}_{4}$N${}_{4}$	AICN	I	7	250	10,700 [92]	3.67	78.5
4-aminoimidazole-5-carboxamide	C${}_{4}$H${}_{6}$N${}_{4}$O	AICA	L		266 [94]	10,700 *	3.67 *	97.8
5-(N’-formamidinyl)-1H-imidazole-4-carbonitrileamidine	C${}_{5}$H${}_{5}$N${}_{5}$	amidine	Am		250	10,700 [95]	6.83 *	80.5 *
adenine	C${}_{5}$H${}_{5}$N${}_{5}$	adenine	A	8	260	15,040 [96]	6.83 [97]	80.5
hypoxanthine	C${}_{5}$H${}_{4}$N${}_{4}$O	hypoxanthine	Hy		250	12,500 [98]	3.16	70.1

Under non-coherent light sources, photochemical reactions can be treated using elementary kinetics equations of the balance type in the product and reactant concentrations. From a detailed analysis of the experiments and calculations performed in the literature, the chemical and photochemical reactions listed in Table 2 will occur in the photochemical dissipative structuring of adenine from HCN. Backward reactions are considered to be negligible except for reaction #9b which is the backward reaction of #9a. The details of each reaction are given after the table.

The following is a detailed description of each reaction given in Table 2 by reaction number:Hydrolysis of hydrogen cyanide HCN (H) gives formamide H${}_{2}$NCOH (F) with a half-life dependent on temperature and pH [68]. The temperature dependent rate equation used here was determined by Kua and Thrush [100] at pH 7.0 from the experimental data of Miyakawa et al. [99].A photon-induced tautomerization converts formamide (F) into formimidic acid (Fa). Basch et al. [90] have measured the electronic excitation spectrum of formamide (F) and find a peak in absorption at 55,000 cm${}^{-1}$ (182 nm) with a molar extinction of 11,000 M${}^{-1}$ cm${}^{-1}$. However, a shoulder exists on the main absorption peak which extends down to 40,000 cm${}^{-1}$ (250 nm). Duvernay et al. [102] suggest that this shoulder arises from the resonant excitation of the forbidden $n\to \pi^{*}$ transition located at 219 nm (130 kcal mol${}^{-1}$) and not from the main $\pi \to \pi^{*}$ transition located at 182 nm. Maier and Endres [101] have determined that irradiation of formamide (F) at 248 nm rapidly converts it into basically two tautomeric isomers of formimidic acid (Fa), H(OH)C=NH, which are both about 3.6 kcal mol${}^{-1}$ in energy above formamide and separated from it by a transition barrier of height of $E_{a}=45.4$ kcal mol${}^{-1}$ (gas phase). Similarly, Duvernay et al. [102] have shown that under UVC light of 240 nm, formamide (F) tautomerizes into formimidic acid (Fa) and their calculation gives a similar transition state barrier height of 47.8 kcal mol${}^{-1}$. Wang et al. calculate a transition state barrier of 49.8 kcal mol${}^{-1}$ [110] but show that this is reduced to 22.6 kcal mol${}^{-1}$ in the presence of only a single water molecule. The energy needed to overcome this barrier is in the infrared (1265 nm) but Cataldo et al. have shown that there is no evidence of thermal excitation until about 220 ${}^{\circ }$C [111]. Our model, therefore, assumes that the F → Fa tautomerization requires the absorption of a photon and we take the wavelength region for tautomerization due to the $n\to \pi^{*}$ transition of 220 ± 10 nm and assign an average molar extinction coefficient to that region of 60 M${}^{-1}$ cm${}^{-1}$ as measured by Basch et al. [90] and also by Petersen et al. [89].Duvernay et al. [102] have shown that formimidic acid (Fa) can, in turn, be photo-lysed into HCN (H) (or HNC) plus H${}_{2}$O, (dehydration) with maximal efficiency at about 198 nm [104]. However, the absorption spectrum of formimidic acid also has a shoulder extending to about 250 nm due to the same $n\to \pi^{*}$ excitation as in formamide. For example, Duvernay et al. observe a small amount of dehydration of formimidic acid at 240 nm. Given that our surface solar spectrum during the Archean (Figure 2) is extinguished below about 205 nm, here we likewise assume an absorption wavelength for photo-lysing of 220±20 nm and a similar average molar extinction coefficient as for the tautomerization of fomamide (F) of 60 M${}^{-1}$ cm${}^{-1}$ which is in accordance with the findings of Gingell et al. [104]. Combining photo-reactions #2 and #3, we thus recuperate some of the HCN lost to thermal hydrolysis as described by reaction #1 (see Figure 6). Barks et al. [103] have shown that if neat formamide is heated (130 ${}^{\circ }$C), thereby exciting vibrational states, a photon-induced excitation at even longer wavelengths (254 nm) also leads to the disintegration of formamide into HCN and H${}_{2}$O, and they believe that this is the route to the production of the purines, adenine, guanine, and hypoxanthine, that they detect. Their yields are increased when including the inorganic catalysts sodium pyrophosphate and calcium carbonate, indicating that heating and inorganic catalysts can improve the photochemical reaction steps #2 and #3. Formamide also disintegrates thermally into HCN and H${}_{2}$O, without requiring the absorption of a photon, but only at temperatures greater than about 220 ${}^{\circ }$C [111] because of high barriers [100].
Figure 6The production of formamidic acid (Fa) from formamide (F) (photoreaction #2) and its subsequent decay into HCN (H) and water (photoreaction #3).
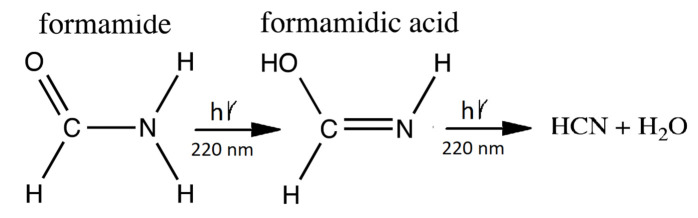
The hydrolysis of formamide (F) gives the salt ammonium formate (Af) at a rate of 1% in 24 h at 100 ∘C [103]. The temperature dependent rate equation given in Table 2 was determined by Kua and Thrush [100] at pH 7.0 from the experimental data of Miyakawa et al. [99]. The ammonium and formate parts of this salt become useful for the thermal reaction leading to the final addition of an HCN (H) to AICN (I) catalyzed by formamide (F) to give adenine (A) (reaction #13) [105,106].HCN (H) thermally polymerizes into (HCN)x with its most stable tetramer (x=4) known as cis-DAMN (C) being the preferred polymer from which more complex polymers can be synthesized [111]. The tetramization of 4HCN is an exothermic thermal reaction and occurs most rapidly at a solvent pH at its pKa value, which decreases with increasing temperature (pKa =8.5 at 60 ∘C and 7.9 at 100 ∘C) [68]. The tetramization of HCN into DAMN is not elementary but involves successive polymerization of HCN with H+ and CN− ions [68] so is second order in the concentration of HCN. The temperature dependence of the rate of conversion of HCN to DAMN has been measured by Sanchez et al. [68]. We assume transition state theory and an Arrhenius equation of form,
(1)k5=exp(−Ea/RT+lnA).From the conversion rates for a 1 M solution of HCN with 0.01 M tetramer catalyst as given in Table 5 of Sanchez et al. [68], a straight line can be fitted to the graph of 1/T vs ln(k5) giving values of lnA=19.049 and Ea/R=9964.3, or Ea=19.8 kcal mol−1. However, this would be the rate equation for tetramerization of HCN at its pKa value which would be about 8.2 at 80 ∘C [68]. To obtain the rate equation at the lower pH value assumed here of 7.0 we note that from Figure 14 of Sanchez et al. [68] the half-lives for tetramization of HCN are the same for pH 7.0 at 80 ∘C as they are for pH 8.2 at 64 ∘C (1 day). Setting the rate constants as the same for these two conditions leads to a value of Ea/R=10,853.01 for pH 7.0. Finally, since HCN can polymerize into either cis-DAMN (C) or trans-DAMN (T), and since trans-DAMN (T) has a free energy of ΔE=0.61 [68] (0.56 [93]) kcal mol−1 higher than cis-DAMN (C) a Boltzmann factor of 1/(exp(−ΔE/RT)+1) is included for cis-DAMN and the same factor but with +ΔE/RT for trans-DAMN.The rates for hydrolysis and polymerization are similar for concentrations of HCN (H) between approximately 0.01 M and 0.1 M (equal rates at 0.03 M for pH 7, T = 80 ∘C, Figure 15 of reference [68]). At lower concentrations, hydrolysis dominates while at higher concentrations polymerization dominates [68].HCN (H) can also thermally polymerize into trans-DAMN (T) which has a free energy of 0.61 kcal mol−1 [68] higher than cis-DAMN (C). We therefore assume that the rate constant for the polymerization into trans-DAMN is the same as that for cis-DAMN multiplied by a temperature dependent Boltzmann factor 1/(exp(+ΔE/RT)+1).Trans-diaminomaleonitrile, trans-DAMN (T), produced through the thermal reaction #6, or through the UV photon-induced transformation of cis-DAMN (C) into trans-DAMN (see reaction #9), is a good catalyst because it has electronic donor parts (–NH2 groups) and acceptor parts (–CN group) linked by a double bond. As such, it can act as a catalyst for the tetramization of 4HCN into cis-DAMN [69]. Cis-DAMN is also a catalyst for the same thermal reactions, but has significantly less activity than trans-DAMN [68] and therefore its catalytic activity is neglected in our analysis. As can be surmised from the discussion of Table 7 of reference [68], including 0.01 M of the tetramer trans-DAMN increases the rate of tetramization by a factor of 12 at 20 ∘C which would correspond to a reduction in the activation energy of 1.45 kcal mol−1. This change in the barrier height is therefore included in the rate constant for this catalyzed reaction.Trans-DAMN also acts as an auto-catalyst for its own thermal production from 4HCN [69] and we assume a similar reduction in barrier height as for its catalysis of the production of cis-DAMN from 4HCN (reaction #7).(a) cis-DAMN can transform into trans-DAMN (3), step (2) →(3) of Figure 4 through a rotation around the double covalent carbon-carbon bond by absorbing a high energy photon (298 nm) to overcome the large energy barrier for rotation, calculated to be 58.03 kcal mol−1 [93] (>4 eV [71]). The quantum yield has been measured by Koch and Rodehorst to be q9=0.045 [92].(b) Because of the high energy barrier 58.03 kcal mol−1 between trans-DAMN and cis-DAMN, thermal energy, even at our high temperatures, is insufficient to significantly reverse the rotation about the double covalent bond. However, trans-DAMN (T) can absorb a photon at 313 nm which would provide it with sufficient energy to isomerize back into cis-DAMN. The quantum efficiency for this reversal was determined to be q9r=0.020 by fitting to the experimental data given in Figure 1 of Koch and Rodehorst [92] as explained in the description of Figure 8.The absorption of a photon at 313 nm excites trans-DAMN which then transforms into AIAC through proton transfer from one of the amino groups [71]. Although there does not appear to exist a quantum yield for this photochemical reaction in the literature, the quantum yield for trans-DAMN to AICN (T→ I) has been measured by Koch and Rodehorst [92] to be 0.0034. By fitting our model concentration results to the experimental data of Koch and Rodehorst (see discussion of Figure 8) it is found that a best value for the quantum yield of trans-DAMN to AIAC (T→J), q10, is 0.006 and therefore the quantum yield for AIAC to AICN (J→ I) would be q11=0.0034/0.006=0.5833.AIAC (J) on absorbing a photon at 275 nm then transforms through photon-induced cyclicization (ring closure) into an azetene intermediate (5 of Figure 4) in an excited state, which then transforms to the N-heterocyclic carbene (6 of Figure 4) and finally this tautomerizes to give the imidazole AICN (I) [71]. As noted above, the quantum yield for this process (J→I) is taken to be 0.5833 to give the overall quantum yield for trans-DAMN to AICN (T→ I) to be 0.0034 [92]. AICN absorbs maximally at wavelength 250 nm.The imidazole, 4-aminoimidazole-5-carbonitrle, AICN (I) created in the previous photochemical reaction (reaction #11) gives the hydrolysis product 4-aminoimidazole-5-carboxamide, AICA (L). The rate equation for this first-order reaction was determined from the data of Sanchez et al. [69] at different temperatures (their Table 1). The barrier to hydrolysis was determined to be 19.93 kcal mol−1 and the frequency factor to be lnA=12.974.The final coupling of a fourth HCN to AICN (I) and its cyclization to form adenine (A) is a very exothermal overall, ΔG=−53.7 kcal mol−1, but there are numerous large energy barriers on the path to its completion [70]. The first step is the coupling of an HCN molecule to AICN, and this appears to be rate limiting since it has the highest energy barrier, calculated in the gas phase, of 39.7 kcal mol−1 [70]. However, it is catalyzed by both bulk solvent and specific water molecules which reduce the barrier to 29.6 kcal mol−1, or by ammonium molecules with bulk water solvent which reduce the barrier further to 27.6 kcal mol−1 [70]. A number of experimental works [103,105,106,112] have revealed that ammonium formate (Af) could provide a route with an even lower barrier, but the rate is still too slow to allow significant adenine production from AICN and ammonium formate, unless a strong concentration mechanism existed, for example, dehydration [103], or perhaps the buildup of concentration inside the vesicle, or the reaction-diffusion self-organizing occurring within the vesicle, as will be considered below.A solution to this rate problem may exist, however, without requiring high concentrations. As early as 1974 Yonemitsu et al. [105] showed that including formamide, the hydrolysis product of HCN (reaction #1), in aqueous solution, or by itself (neat solution), along with ammonium formate could dramatically speed up the reaction as long as the temperature was above approximately 80 ∘C, leading to a successful industrial patent for the production of adenine from cis-DAMN (C) or trans-DAMN (T) and formamide with ammonium formate. From examples 1 and 12 of the experiments of Yonemitsu et al. carried out at 150 and 100 ∘C (using 135 g of formamide, 30 g of ammonium formate, and 2.01 g of DAMN) giving rise to 43.5% and 30.0% product of adenine after 5 and 10 h at those temperatures respectively, it is possible to calculate an activation barrier for the overall reaction of Ea=6.682 kcal mol−1. Since ammonium formate is a salt, the probable pathway from AICN to adenine would be that proposed by Zubay and Mui [106] where the ammonium ion NH4+ attacks the triple NC bond of AICN and the formate ion HCOO− attacks the amine NH2 group of AICN (Figure 8 of reference [106]) both catalyzed by the proton transfer process involving formamide (see below), leading to this very low barrier. We therefore assume the reaction to be of second order and determined by the Arrhenius equation of form,
(2)k13=exp(−Ea/RT+lnA),
where Ea=6.682 kcal mol−1 and the pre-exponential frequency factor A was estimated from the reduced mass dependence of the Langevin model [113], A=2πeα/μ for a charged ion - neutral molecule system where *e* is the ion electronic charge, α is the polarizability of the neutral reactant, and μ is the reduced mass of the reactants [114]. Considering all factors being equal except the reduced mass, and then normalizing to the frequency factor of reaction #12 for the hydrolysis of AICN (I) by the inverse square root of the reduced mass for the reacting species, gives a value of lnA=12.9734.There exists a second possible route to adenine from AICN and HCN, without involving ammonium formate but considering the catalytic effect of formamide. Wang et al. [107] have studied, through ab initio DFT, the synthesis of adenine starting from pure formamide and propose what they call a “formamide self-catalytic mechanism”. This mechanism consists of; (1) a proton transfer from N to O of formamide to form the imidic acid tautomer, formimidic acid (Fa), potentially obtained in our case through a photon-induced proton transfer reaction #2; (2) a proton exchange between one imidic tautomer and one amide tautomer, resulting in two formimidic acids; and; (3) an interaction between these two imidic acids yielding formimidic acid, a water molecule, and HCN. This formamide self-catalytic mechanism has relevance to the entire adenine synthesis process starting from pure formamide since it reduces many of the barriers on route to adenine [107].Of importance to us here of Wang et al.’s results is the step of the attachment of HCN to the amine group (NH2) of AICN. They show for their particular case of formiminylation of 5-aminoimidazole (Figure 13 of reference [107]) that this reaction can be formamide-catalyzed (as described above) and find the activation energy barrier for this to be 19.9 kcal mol−1 (significantly lower than 46.1 kcal mol−1 in the noncatalyzed process and 34.0 kcal mol−1 in the water-assisted process) and that the subsequent dehydration process to give the amidine (Am) (our case) is calculated to be 14.0 kcal mol−1 (34.3 kcal mol−1 in the noncatalyzed reaction).Therefore, we assume that the attachment of HCN (H) to AICN (I) to form 5-(N’-formamidinyl)-1H-imidazole-4-carbonitrileamidine (Am) to be a formamide-catalyzed thermal reaction involving formimidic acid and formamide and we assume the rate of this reaction to be determined by the Arrhenius equation of form
(3)k14=exp(−Ea/RT+lnA)
where Ea=19.9 kcal mol−1 and the pre-exponential frequency factor A is again estimated from the reduced mass dependence of the Langevin model [113], considering again all factors equal except the reduced mass, and then normalizing to the reaction #12 for the hydrolysis of AICN (I) by the inverse square root of the reduced mass for the reacting species, giving a value of lnA=12.613.Note that AICN (I) has a conical intersection for a charge transfer from the molecule in the excited state to a neighboring cluster of water molecules [115]. With AICN left in the charged state, this would significantly increase the rate of attachment, through charge-dipole interaction, to formamide, which has a dipole moment significantly larger than that of water (Table 1), effectively changing the reaction from third order to second order, thereby significantly increasing the overall rate of this last attachment of HCN to AICN through this formamide-catalyzed reaction.The possibility of a hot ground state reaction occurring to aid in overcoming the barrier to producing adenine (A) from AICN (I) and HCN (H) could also be considered during daylight periods. These occur within a narrow time window after photon excitation, calculated by Boulanger et al. for a molecule (trans-DAMN) which has a similar conical intersection as AICN, to be about 0.2 ps, which corresponds to the time at which the excess energy on the molecule has been reduced to about 1/3 of its initial value, allowing reactions to proceed with a maximum barrier height of about 30 kcal mol−1 [71]. This possibility, however, is not included in the model. It would have the overall effect of increasing the rate of the production of adenine.After the attachment of a fifth HCN (H) to AICN (I) to form the amidine (Am), reaction #14, a subsequent tautomerization is required (calculated to have a high barrier of about 50 kcal mol−1) which, once overcome, allows the system to proceed through a subsequent barrier-less cyclicization to form adenine [95]. Such a high barrier to the final cyclicization means that, at the temperatures considered here, it cannot be a thermal reaction. Indeed, the fact that adenine has been found in space and in meteorites where temperatures are expected to be very low, indicated to Glaser et al. [95] that a photochemical route must be available. They suggested a photon-induced tautomerization of amidine, which absorbs strongly at 250 nm. Although oscillator strengths for the tautomerization have been calculated by Glaser et al., different ab initio approaches give significantly different values, so experiment will be required for its reliable determination. Therefore, until such data becomes available, we assume a similar molar extinction coefficient as for AICN and, being conservative, a quantum efficiency of q15=0.06 but measure the effect on adenine production for a ±30% variation of this parameter value (see Table 4). In fact, the results of Table 4 indicate that due to the large activation energy required, and the fact that the reactions are of second order, reactions #14 and #15 only come into play at very high temperature.The temperature dependent rate equation for the destruction of adenine (A) through hydrolysis to give hypoxathine (Hy) which could then lead to guanine, or through deamination to some amino acids [116], was determined in careful experiments by Levy and Miller [108] (and by Wang and Hu [109]). Zheng and Meng calculated a transition state barrier for hydrolysis of 23.4 kcal mol−1 [117].to 24. These reactions represent the absorption of a photon, in a 20 nm region centered on the wavelength of peak absorption, on the molecule which then decays through internal conversion at a conical intersection to the ground state on sub-picosecond time scales. All molecules listed in this set of photo-reactions are basically photo-stable because of a peaked conical intersection connecting the excited state with the ground state. These reactions, with large quantum efficiencies, represent the bulk of the flow of energy from the incident UVC spectrum to the emitted outgoing ocean surface spectrum in the infrared and therefore contribute most to photon dissipation, or entropy production.

To obtain simple kinetic equations for the photochemical reactions listed in Table 2, the molecules are assumed to only absorb within a region ±10 nm of their maximum absorption wavelength $\lambda_{max}$ and that this absorption is at their maximum molar extinction with coefficient ϵ (Table 1), and finally that there is no shadowing in overlapping absorption wavelength regions. It is assumed that a 100 μm diameter vesicle is at the ocean surface and the depth coordinate is divided into i=20 bins of width Δx=5μm and the time interval for the recursion calculation for the concentrations at a particular depth is 10 ms. Results for the model are relatively independent of vesicle diameter.

The recursion relation for the factor of light intensity $L_{\lambda }\left(i,C\right)$ for a concentration *C* of the molecule, at a depth x(i)=i·Δx below the ocean surface will be,
(4)$$L_{\lambda }\left(i,C\left(i\right)\right)=L_{\lambda }\left(i-1,C\left(i-1\right)\right)e^{-\Delta x\cdot \alpha_{\lambda }}\cdot 10^{-\Delta x\cdot \epsilon_{\lambda }C\left(i\right)}$$
where $\alpha_{\lambda }$ is the absorption coefficient of water at wavelength λ and $\epsilon_{\lambda }$ is the molar extinction coefficient of the particular absorbing substance which has concentration C(i) at x(i).

The kinetic equations giving the increment in concentration after each time step Δt≡dt, for use in a discrete recursion relation, at a depth *x* below the surface, are determined from the reactions listed in Table 2 to be:
(5)dHdt=DH∂2H∂x2−k1H+d·q3I220L220(Fa)(1−10−Δxϵ220Fa)Δx−k5H2−k6H2−k7H2T−k8H2T=DH∂2H∂x2+d·q3I220L220(Fa)(1−10−Δxϵ220Fa)Δx−Hk1−H2(k5+k6+T(k7+k8))(6)dFdt=DF∂2F∂x2+k1H−d·q2I220L220(F)(1−10−Δxϵ220F)Δx−k4F−k14IFa(7)dFadt=DFa∂2Fa∂x2+d·q2I220L220(F)(1−10−Δxϵ220F)Δx−d·q3I220L220(Fa)(1−10−Δxϵ220Fa)Δx(8)dAfdt=DAf∂2Af∂x2+k4F−k13IAf(9)dCdt=DC∂2C∂x2+k5H2+k7H2T−d·q9I298L298(C)(1−10−Δxϵ298C)Δx+d·q9rI313L313(T)(1−10−Δxϵ313T)Δx(10)dTdt=DT∂2T∂x2+k6H2+k8H2T+d·q9I298L298(C)(1−10−Δxϵ298C)Δx−d·q10I313L313(T)(1−10−Δxϵ313T)Δx−d·q9rI313L313(T)(1−10−Δxϵ313T)Δx(11)dJdt=DJ∂2J∂x2+d·q10I313L313(T)(1−10−Δxϵ313T)Δx−d·q11I275L275(J)(1−10−Δxϵ275J)Δx(12)dIdt=DI∂2I∂x2+d·q11I275L275(J)(1−10−Δxϵ275J)Δx−k12I−k13IAf−k14IFa(13)dLdt=DL∂2L∂x2+k12I(14)dAmdt=DAm∂2Am∂x2+k14IFa−d·q15I250L250(Am)(1−10−Δxϵ250Am)Δx(15)dAdt=DA∂2A∂x2+d·q15I250L250(Am)(1−10−Δxϵ250Am)Δx+k13IAf−k16A(16)dHydt=DHy∂2Hy∂x2+k16A
where the differentials are calculated discretely (e.g., dH/dt≡ΔH/Δt) and all concentration values are calculated at discrete time steps of Δt=10 ms and the calculated value of the change (e.g., ΔH(j)/Δt) for time step *j* is summed to the previous value (e.g., H(j−1)). The first terms on the right of the equal sign represent diffusion flow, with $D_{Y}$ the diffusion constant of molecule *Y*. The day/night factor *d* is equal to 1 during the day and 0 at night. $I_{220},I_{298},I_{313},I_{275}$ and $I_{250}$ are the intensities of the photon fluxes at 220,298,313,275 and 250 nm respectively (Figure 2). $\epsilon_{\lambda }$ are the coefficients of molar extinction for the relevant molecule at the corresponding photon wavelengths λ.

### 4.3. Vesicle Permeability and Internal Diffusion

The permeability of the vesicle wall to the molecule, and the diffusion constant for the molecule within the aqueous region of the interior of the vesicle will both decrease with the area of the molecule and with the size of its electric dipole moment (Table 1) and increase with temperature. It is interesting to note that almost all the final and intermediate product molecules have large dipole moments, implying tendency towards entrapment within the vesicle. We assume that the vesicle cannot remain intact at temperatures greater than 95 ${}^{\circ }$C but that below this temperature it is completely permeable to H${}_{2}$O, HCN (H) and formimidic acid (Fa) but impermeable to all the other intermediate products due to their large size and large electric dipole moments. Ammonium formate would be in its ionic form and therefore also unable to cross the fatty acid membrane since permeability across bi-lipid membranes is reduced by orders of magnitude if the molecules are polar or charged [118].

The diffusion constant D${}_{Y}$ for the molecule *Y* will depend on the viscosity of the solution inside the vesicle, which is dependent on the amount of organic material within the vesicle. Studies of intracellular diffusion of nucleotides indicate three factors influencing diffusion rates besides temperature at high solute densities; the viscosity of the medium, collisional interactions dependent on concentration, and binding interactions between molecules [119]. The diffusion constant of adenine in pure water has been determined to be $D_{A}=7.2\times 10^{-6}$ cm${}^{2}$ s${}^{-1}$ [120] while the measured diffusion rates in the cytoplasm of different cell types varies between $1.36\times 10^{-6}$ to $7.8\times 10^{-6}$ cm${}^{2}$ s${}^{-1}$ [119].

Surface films of organics and trace metals, with a high density of lipids and other hydrocarbons, produced for example by the ultraviolet spectrum of Figure 2 on CO${}_{2}$ saturated water [17], could have been expected on the ocean surface during the Archean. Diffusion constants in this sea surface microlayer would then be significantly smaller than for the bulk water. Diffusion rates inside the vesicle will depend on the amount of organic material already existing at the air/water interface (this may have varied spatially considerably) captured during the formation of the vesicle, and on the amount of ongoing organic synthesis within the vesicle.

All diffusion constants are defined relative to that for adenine through the formula;
(17)$$D_{Y}=\frac{\mu_{A}A_{A}}{\mu_{Y}A_{Y}}\cdot D_{A},$$
where $A_{A}$ is the polar surface area and $\mu_{A}$ the dipole moment of adenine (Table 1). Here we investigate two different diffusion constants, the smallest value for adenine in present-day cytoplasm and one four orders of magnitude smaller. Using Equation (Equation 17) and the values given in Table 1 for the molecule dipole moment and polar area, we obtain the results given in Table 3.

A more refined model could consider dynamic diffusion (as a function of the molecular concentration increases inside the vesicle) and a more individualized molecular membrane permeability, allowing leakage of the products into the surrounding water when internal concentrations become large. Such vesicles could be considered as factories, seeding the ocean microlayer with UVC pigments (fundamental molecules), and making the initial conditions more favorable for further evolution through dissipative structuring in later vesicles.

Cyclical boundary conditions are assumed for diffusion, except for HCN (H) and formimidic acid (Fa) which can permeate the vesicle wall and therefore at the wall they are given their fixed value specified in the initial conditions (see following subsection) assumed for the environment outside the vesicle. The second order derivatives for calculating the diffusion were obtained using the second order finite difference method with double precision variables.

### 4.4. Initial Conditions

Miyakawa, Cleaves and Miller [99] estimated the steady state bulk ocean concentration of HCN at the origin of life assuming production through electric discharge on atmospheric methane to produce radicals which attack N${}_{2}$, leading to an input rate to the oceans of 100 nmole cm${}^{-2}$ y${}^{-1}$, and loss of HCN due to hydrolysis and destruction at submarine vents with a 10 million year recycling time of all ocean water for an ocean of 3 Km average depth. For an ocean of pH 6.5 and temperature of 80 ${}^{\circ }$C, they obtained a value of [HCN] $=1.0\times 10^{-10}$ M [99].

However, as mentioned above, HCN can also be produced through the solar Lyman alpha line (121.6 nm) photo-lysing N${}_{2}$ in the upper atmosphere giving atomic nitrogen which then combines with CH and CH${}_{2}$ to give HCN, or through 145 nm photolysis of CH${}_{4}$ leading to a CH^*^ radical which attacks N${}_{2}$ to give HCN [60]. Including this UV production would increase the input of HCN to the oceans by a factor of at least 6 [3,72,121]. Furthermore, the first ∼100 μm of the ocean surface, the hydrodynamic boundary layer, is now known to be a unique region in which surface tension leads to enriched organics with densities up to $10^{4}$ times that of organic material in the water column slightly below [77]. Trace metal enhancement in this microlayer can be one to three orders of magnitude greater than in the bulk [77,122]. Langmuir circulation, Eddy currents, and the scavenging action of bubbles tends to concentrate organic materials into this surface film. If disturbed or mixed, the film rapidly reestablishes its integrity. This high density of organic material trapped through hydrophobic and ionic interactions at the ocean surface leads to significantly lower rates of diffusion at the surface microlayer as compared to the ocean bulk [77]. Little diffusion and turbulence therefore imply little mixing. The ocean microlayer is therefore a very stable layer which would also not be recycled through ocean vents. Finally, although HCN is very soluble in bulk water, recent molecular dynamic simulations have shown that it concentrates to about an order of magnitude larger at the air-water interface due to lateral HCN dipole-dipole interactions, and that it evaporates at lower rates than does water [79].

Therefore, rather than assuming the low bulk concentrations of Miyakawa et al. [99], we instead consider two higher initial surface concentrations for HCN (H) ($6\times 10^{-5}$ and $6\times 10^{-4}$ M) and formimidic acid (Fa) ($1\times 10^{-5}$ and $1\times 10^{-4}$ M), the latter resulting from a photochemical tautomerization of formamide, the hydrolysis product of HCN (reactions #1 and #2 of Table 2). We also allow for the perturbation of the system by considering the probable existence of small and sparse patches of much higher concentrations, up to 0.1 M, of both these molecules, justified by the above-mentioned characteristics of the ocean microlayer and the dipole-dipole interaction between HCN molecules. The initial concentrations of all other reactants and products inside the vesicle (assumed impermeable to these) are taken to be $1.0\times 10^{-10}$ M.

There is significant uncertainty in the date of the origin of life and in the temperature of Earth’s surface at that time. There is a consensus, however, that it occurred after “the late lunar bombardment” at ∼3.9 Ga in either thermal or hyperthermal conditions, with Earth’s surface cooling throughout the Archean. It is therefore relevant to consider whether the dissipative structuring of adenine under UVC light proposed here could have been efficient over a range of temperatures (60–95 ${}^{\circ }$C), including, perhaps, conditions which may have existed in the Hadean before the putative event of the origin of life.

## 5. Results

### 5.1. Validation of Model

The rates of tetramization of 4HCN (H) to cis-DAMN (C) and trans-DAMN (T) are given by the terms $k_{5}H^{2}$ and $k_{6}H^{2}$ of Equations () and (), reactions #5 and #6, respectively. The rate of hydrolysis of HCN (H) into formamide (F) is given by the term $k_{1}H$ of Equation (), reaction #1. The ratio of these rates $H^{2}\left(k_{5}+k_{6}\right)/Hk_{1}$ for pH 7.0 at different concentrations of HCN and as a function of temperature is plotted in Figure 7 along with experimental values derived from the data of Sanchez et al. [68] for the point at which tetramization and hydrolysis rates are equal (ratio =1).

Figure 7 shows that ratio of the rates of HCN tetramization to hydrolysis increases with HCN concentration and with lower temperature. For this reason, eutectic concentration at freezing temperatures was deemed to be the most probable route from HCN to the nucleobases, giving rise to the “cold origin of life” scenarios [74,75,76]. However, irrespective of the fact that this contradicts the available geochemical evidence of high temperatures during the Archean, it will be shown here that high temperatures could also have led to significant concentrations of the nucleobases for the following reasons, (1) the ocean surface microlayer is a region of orders of magnitude higher organic density than the bulk, (2) UVC photochemistry on HCN inside a fatty acid vesicle would allow a buildup of those product molecules unable to permeate the vesicle wall, (3) hydrolysis of HCN leads to formamide (F) (reaction #1), and a subsequent hydrolysis to ammonium formate (Af) (reaction #4), the former of which is an important catalyst, and the latter a necessary component, for the final attachment of a 5th HCN molecule to AICN (I) to give adenine (reaction #13) which occurs with great efficacy above temperatures of 80 ${}^{\circ }$C [105], and (4) an alternative route to adenine is reaction #14 which because of its high activation energy would occur only at very high temperatures (>95 ${}^{\circ }$C).

Experiments have been performed by Koch and Rodehorst [92] concerning the UV photo-transmutation of cis-DAMN (C) into trans-DAMN (T) and then into AICN (I) (Figure 1 of reference [92]) which are the important photochemical steps in our model. This occurs through three photochemical reactions $\gamma_{298}+C\to T$, $\gamma_{313}+T\to J$, $\gamma_{275}+J\to I$, where “I” is AICN and the intermediate ”J” is AIAC (Figure 4). Our model can be compared to these experimental results since the light source used by Koch and Rodehorst was stipulated as being a Rayonet RPR3000 A lamp which peaks in intensity at 305 nm with ∼10% smaller and similar output at both 313 and 298 nm, and about 10% of the latter at 275 nm (see Figure 13 of reference [123]). These ratios of Rayonet RPR3000 A lamp light intensity at 313:298:275 nm of 1.0:1.0:0.1 were used in our model and all initial concentrations set to zero except that of cis-DAMN (C), which was set to 0.00145 M (Figure 1 of reference [92]). The temperature was set to the 20${}^{\circ }$ of experiment. The day/night light cycling was disabled and the two quantum efficiencies, unavailable in the literature, for $\gamma_{313}+T\to J$ and $\gamma_{313}+T\to C$, were adjusted to $q_{10}=0.006$ and $q_{9r}=0.020$ to give a best fit of the model to the experimental data. Determining $q_{10}$ in this manner then determines $q_{11}$ since $q_{10}\times q_{11}=0.0034$ [92]. The overall intensity of the light on sample was adjusted to give the correct time scale. The results, plotted in Figure 8, indicate that our full model, employing the initial conditions of experiment, can reproduce well the shapes of the three experimental data sets by fitting with only two parameters, the quantum efficiencies $q_{9r}$ and $q_{10}$.

Further validation of our model comes from the fact that at the photostationary state under the Rayonet lamp, Koch and Rodehorst find that the remaining DAMN is distributed between its two isomers trans (T) and cis (C) with proportions of 80% and 20% respectively [92]. Our model at close to the stationary state, at 6800 min (Figure 8), gives these proportions as 77% and 23% respectively.

Using instead the UV light intensities of the Archean surface UV solar spectrum (Figure 2) gives the time dependent concentration profiles as shown in Figure 9. The difference between Figure 8 and Figure 9 are due to the differences in the incident light spectra, principally the light intensity at 298 nm (responsible for the C →T isomerization). The intensity at this wavelength in the solar spectrum arriving at the Archean Earth surface was an order of magnitude smaller than that of the Rayonet lamp used in the experiments.

The catalytic effect of trans-DAMN on the tetramization of HCN (reaction #7) was incorporated into our model by reducing the energy of the activation barrier such as to give the same amplification factor of 12 due to the catalytic effect of the inclusion of 0.01 M trans-DAMN in the HCN solution observed in the experiments of Sanchez et al. [68] at a temperature 20 ${}^{\circ }$C (see discussion of reaction #7 after Table 2).

All other parameters employed in the model, such as activation barrier energies, pre-exponential frequency factors, and quantum efficiencies (except $q_{15}$), were taken directly from experiment, or by fitting to experimental rate versus temperature data, or taken from accurate first principles calculations as described in the list of reaction details found after Table 2. However, to determine the sensitivity of our model results to possible inaccuracies in the parameters, in Table 4 the critical parameters of the model (those quantum efficiencies not determined directly by experiment, or large activation energies) are varied by ±30%, and the effect on the final adenine concentration is noted after 30 Archean days at 80 ${}^{\circ }$C.

From Table 4 it can be seen that at 80 ${}^{\circ }$C the parameter variations with greatest effect on the concentration of adenine are, as expected, the first-order hydrolysis reactions, #12 for hydrolysis of AICN and #16 for hydrolysis of adenine itself. Reducing the nominal activation barrier for adenine loss through hydrolysis determined by Levy and Miller [108] (half-life of adenine of 8.0 years at 80 ${}^{\circ }$C at neutral pH) by 30% leads to an almost 4 order of magnitude decrease in the final concentration of adenine after 30 days as compared to its final concentration using the nominal hydrolysis activation energy. It is noted that the hydrolysis of adenine leads to guanine, or, through deamination, to some amino acids [116], so its occurrence at some rate would have been important to the origin of life. Changing the parameters for reactions #14 and #15 does not affect adenine production because this route to adenine through amidine (Am) only comes into play at temperatures above ∼95 ${}^{\circ }$C because of the high activation energy and the fact that reaction #14 is of second order. Most of the adenine production at 80 ${}^{\circ }$C occurs through reaction #13 and variation of this activation energy $E_{a13}$ has little effect on the final concentration of adenine since the activation energy is low.

### 5.2. Evolution of the Concentration Profile

Figure 10, Figure 11, Figure 12, Figure 13 and Figure 14 present the time evolution in Archean days (16 h) of the concentrations of the relevant molecules in the photochemical synthesis of adenine inside the vesicle obtained by solving simultaneously the differential kinetic Equations (5)–(16), for the initial conditions and diffusion constants listed in the figure captions.

The concentration profiles of the molecules evolve over time because of the accumulation of photoproducts within the vesicle. A deliberate external perturbation, effectuated at 10.4 Archean days, of the non-linear system leads it to a new stationary state in which the environmental precursor molecule HCN is converted into adenine at a much greater rate. This leads to greater dissipative efficacy of the system, i.e., to a concentration profile of the molecules which dissipates more efficiently the incident UVC spectrum.

The dynamics observed in the concentrations profiles displayed in Figure 10 and Figure 11 is a function of both the external perturbations affecting the system and of its inherent non-linearity. Given the fixed concentrations of HCN (H) and formimidic acid (Fa) in the environment, to which the vesicle is permeable, photochemical reactions occur during daylight hours (denoted by the violet-colored sections of the horizontal dashed line labeled as D/N). This gives rise to the observable diurnal oscillations in the concentrations of trans-DAMN (T) and AICN (I) since these are direct products of photochemical reactions.

At 10.4 Archean days, the vesicle is perturbed by assuming it passes through a region of high density of HCN (H) and formimidic acid (Fa) (0.1 M) for a 2 min period (vertical blue line at top of (a) graphs). This sudden impulse in HCN and Fa concentration gives rise to rapid increases in all concentrations within the vesicle, in particular for formamide (F), the hydrolysis product of H, which is an important catalyst for reaction #13 which produces adenine (A) from AICN (I) (see Table 4) and this reaction route is the most important for adenine production due to its low activation barrier. Ammonium formate (Af) is used up in this reaction so its concentration decreases after the perturbation. More importantly, however, immediately after the perturbation there is a greater production of trans-DAMN (T) in the vesicle and since T acts as a catalyst for the polymerization of HCN (H) (reactions # 7 and #8), this will produce a greater metabolism of H into DAMN within the vesicle and therefore a stronger diffusion of H into the vesicle from the outside environment as long as T remains higher than before the perturbation. In other words, the reason that a short impulse of HCN and formimidic acid gives rise to an important increase in the rate of production of adenine is that the vesicle’s semi-permeable wall, together with the set of equations describing the photochemical and chemical reactions, Equations (5)–(16), form a non-linear system with more than one stationary state solution available at any given time.

Perturbation causes the system to leave the attraction basin of one solution determined by its initial conditions and evolve towards a different, and more probable, stationary state of much higher rate of production of adenine (given by the slope of the black trace of Figure 11, more obvious when plotted on a linear scale, Figure 12). The second stationary state is more probable than the initial under the UVC light since the forward reactions are more probable than the reverse (Figure 3). Its concentration profile is more dissipative, i.e., with more molecules having conical intersections to internal conversion rather than to other photochemical products. The thermodynamic driving force for this evolution to the new stationary state after the perturbation is, therefore, greater photon dissipative efficacy. For non-equilibrium systems where *local equilibrium* (Appendix A) is valid, this is the same as saying that the entropy production of the system increases (see Figure 18). This is an example of macroscopic dissipative structuring, in this case of the concentration profile, and this is the physics and chemistry behind biological evolution at its earliest stages.

Figure 13 shows the results obtained by increasing the concentrations of HCN (H) and formimidic acid (Fa) in the environment by an order of magnitude to $6\times 10^{-4}$ and $1\times 10^{-4}$ M, respectively. Comparing the adenine production in Figure 11a, obtained with a single perturbation of H and Fa to 0.1 M for two minutes with an environmental concentration of H of $6\times 10^{-5}$ M, with Figure 13b without perturbation but an environmental concentration of H of 10 times higher at $6\times 10^{-4}$ M, emphasizes the fact that it is not the cumulative exposure of the vesicle to environmental HCN (H) concentration that most effects the rate of production of adenine, but rather the non-linearity which allows a large perturbation to lead the system into a new production regime (new stationary state).

Figure 14 shows the results obtained with a diffusion exponential four orders of magnitude smaller, at $1.0\times 10^{-10}$ cm${}^{2}$ s${}^{-1}$. The small diffusion constant allows the observation of the coupling of the reactions with diffusion, leading to spatial symmetry breaking of the concentration profiles (Figure 15 and Figure 16).

Figure 15 and Figure 16 plot the product concentration profiles as a function of depth below the ocean surface for the initial conditions of Figure 14 at the time of 10.7 Archean days (5 h after the perturbation). The coupling of reaction to diffusion leads to a non-homogeneous distribution of products within the vesicle, with a general increase in concentration towards the center of the vesicle. Stationary state coupling of reactions to diffusion, leading to particular regions of high concentration of the products, was first shown to occur for purely thermal reactions with different activator and inhibitor diffusion rates by Turing [124] and studied more generally as dissipative structures under the framework of CIT theory by Glansdorff and Prigogine [125]. This spatial symmetry breaking is another form of macroscopic dissipative structuring which could facilitate a subsequent UVC polymerization of nucleobases into oligos (assuming UVC-assisted synthesis of ribose from similar precursor molecules [62] and either a high temperature [126], or formamide, catalyzed [127] phosphorylation—not considered here). The tendency days after the perturbation is towards homogeneity. Without perturbation, the concentration profiles remain homogeneous over the vesicle.

The temperature dependence of the amount of product molecules obtained after 30 Archean days is given in Figure 17. Ammonium formate (Af) is produced by the hydrolysis of first HCN (H) to formamide (F) (reaction #1) and then hydrolysis of formamide to Af (reaction #4). Both of these reactions have high activation energies, and this results in Af only being produced in significant quantities at temperatures greater than 80 ${}^{\circ }$C. Most of the adenine (A) production occurs through reaction #13 which consumes Af and therefore high temperatures are important to the production of adenine.

It is instructive to compare our overall non-equilibrium results obtained with the model of UVC production of adenine from HCN within a lipid vesicle with the quasi-equilibrium experiments performed by Ferris et al. [94]. Starting with a high 0.1 M concentration of HCN in water (pH 9.2), and allowing this solution to polymerize in the dark at room temperature for 7 months, and then subjecting these polymers to hydrolysis at 110 ${}^{\circ }$C for 24 h, Ferris et al. obtain an adenine yield of 1 mg l${}^{-1}$ (equivalent to a concentration of $7.4\times 10^{-6}$ M - the molar mass of adenine being 135.13 g mol${}^{-1}$). Our model gives a similar adenine concentration of $7.3\times 10^{-6}$ M within 30 days (Figure 11), starting from a much lower and more realistic initial concentration of HCN of only $6.0\times 10^{-5}$ M (including one perturbation of HCN concentration to 0.1 M for only two minutes) and a more natural neutral pH of 7.0 at 80 ${}^{\circ }$C and under a UVC flux integrated from 210–280 nm of about 4 W m${}^{-2}$ during daylight hours (Figure 2). At 90 ${}^{\circ }$C, under the same conditions, adenine concentration more than doubles to $1.7\times 10^{-5}$ M (Figure 10).

In Figure 18a I plot the entropy production as a function of time in Archean days due to the photon dissipation by the corresponding molecular concentration profile, as represented by reactions 17 to 24 of Table 2. In general, the entropy production is an increasing function of time. These photo-reactions represent the terms $d_{J}P/dt$ of Appendix Equation (Equation 19), and even though the terms $d_{X}P/dt$ which represent the variation of the entropy production due to rearrangement of the chemical affinities (the free forces *X*), are negative definite (corresponding to the structuring of the molecules) consistent with the Glansdorf–Prigogine universal evolutionary criterion (Appendix A), the total entropy production $dP/dt=d_{J}P/dt+d_{X}P/dt$ increases. This is due to the fact that $d_{J}P/dt$ represents the entropy production due to the chemical/photochemical reactions plus the contribution due to the dissipation of the photons which are flowing through the system and being converted from short wavelength UV into long wavelength infrared (dissipated) light, and this photon flow captured by the system increases over the evolution of the concentration profile of the intermediate products within the vesicle, particularly after the 2 min perturbation of the system at 10.4 Archean days. In Figure 18b I plot the same entropy production but for the case in which there is no perturbation of the system (the environmental concentrations of HCN and formimidic acid (Fa) are kept constant at $6\times 10^{-5}$ and $1\times 10^{-5}$ M respectively). In this latter case, the entropy production remains almost 3 orders of magnitude lower.

## 6. Discussion

Not only do high ocean surface temperatures promote the buildup of product concentration (Figure 17) but they would also foment phosphorylation with phosphate salts and formamide, favoring the formation of acyclonucleosides and the phosphorylation and trans-phosphorylation of nucleosides which only occurs efficiently at temperatures above 70 ${}^{\circ }$C [126,127].

Besides the UVC photochemical buildup and entrapment of product molecules inside the vesicle, another concentration mechanism for this system arises through the coupling between reaction and diffusion in the non-linear regime which leads to the breaking of spatial symmetry (e.g., the Belousov–Zhabotinsky reaction [37]). For low diffusion rates, the homogeneous stationary state is no longer stable with respect to a space dependent perturbation and intermediate products become preferentially concentrated at the center. The importance of this increases with lower diffusion rates (Figure 14, Figure 15 and Figure 16).

The other important purine of RNA and DNA, guanine, can be produced from AICA (L) (the hydrolysis product of AICN, reaction #12 of Table 2), through a thermal reaction with either cyanogen (CN)${}_{2}$ or cyanate (OCN${}^{-}$). Cyanogen can be generated from HCN (H) either photochemically [128] or thermally [129]; cyanate is obtained from cyanogen through hydrolysis [68]. The production of guanine from AICA would increase the photon dissipation of the system, as can be surmised by comparing the molar extinction coefficients and wavelengths of maximum absorption of these two molecules, and therefore the concentration of guanine would increase, or, in other words, be *dissipatively selected* by the same mechanism (Figure 3) and the non-linear, non-equilibrium, thermodynamics allowing perturbations to take the system to new stationary states of greater purine production rates and greater photon dissipation, as explained in the previous section and in Appendix A.

Regarding the pyrimidines cytosine, uracil, and thymine, Ferris, Sanchez and Orgel [130] showed that on heating to 100 ${}^{\circ }$C a 5:1 ratio of cyanate with cyanoacetylene, cytosine was formed in yields of 19%. In this reaction, cytosine is formed mainly in a sequence involving the stable intermediate cyanovinylurea. Cyanogen or cyanoformamide can replace cyanate in this synthesis. The hydrolysis of cytosine readily gives uracil, and when uracil is reacted with formic acid in dilute aqueous solutions at 100–140 ${}^{\circ }$C, thymine is formed [131]. All the purines and pyrimidines can therefore be obtained within the same non-equilibrium non-linear vesicle model by assuming only HCN and some acetylene (C${}_{2}$H${}_{2}$) dissolved in a water solvent at high temperature under UVC light.

Inorganic catalysts were not included in our model but can increase the rate of purine production. For example, Cu${}^{+2}$ ions have a large effect in increasing the rate constant for the conversion of HCN (H) to cis-DAMN (C) [68] (reaction # 5). Cu${}^{+2}$ ions also reduce the energy difference (but not the barrier crossing height) between the isomers formimidic and formamidic acid of formamide [132]. Metal ions would have been in high abundance at the ocean surface microlayer [77,78].

## 7. Summary and Conclusions

Understanding the origin of life requires the delineation of a coherent physical-chemical framework for the various continuous and sustained dissipative processes involved; synthesis, proliferation, and evolution towards complexes of greater dissipation. Early life appears to have been a particular form of non-equilibrium dissipative structuring leading to ever greater UVC photon dissipative efficacy; *microscopic* dissipative structuring of carbon-based pigment molecules under Archean UVC light. The synthesized pigments absorbed strongly in the UVC region and were endowed with peaked conical intersections allowing the efficient dissipation of this absorbed light into heat. These dissipative structures attain stability once endowed with a conical intersection to internal conversion since this reduces the quantum efficiency for deexcitation through further photochemical reaction pathways. It was, however, not a fortuitous coincidence, nor a requirement for photo-stability, that the fundamental molecules of life have these photochemical characteristics (Figure 2) but rather that these characteristics are, in fact, the “design” goals of dissipative structuring. Selection in nature is effectively based on the dissipative efficacy of the structure (Figure 3).

The initial dissipative structuring at the origin of life must necessarily have occurred in the long wavelength UVC region where there was enough energy to directly break and reform carbon double covalent bonds while not enough energy to disassociate these molecules through successive ionization. Photochemical reactions in this wavelength region provide a much richer suite of microscopic mechanisms for dissipative structuring than do thermal chemical reactions. These mechanisms include tautomerizations, disassociations, radicalizations, isomerizations, charge transfers, additions, and substitutions.

Such molecules with peaked conical intersections and presenting broad absorption would then form a basis set of molecules (fundamental molecules) for the subsequent construction of all early dissipative structures and processes of life. Unlike macroscopic dissipative structures such as hurricanes or convection cells, at normal temperatures these microscopic dissipative structures remain intact even after the removal of the imposed light potential responsible for their synthesis due to strong covalent bonding between atoms. The corresponding macroscopic dissipative structures are the concentration profiles of these molecules and these profiles can also evolve towards profiles of greater photon dissipative efficacy because of the non-linearity in the reaction set giving rise to multiple stationary states, each with different dissipative efficacy.

As an example, I presented a simple kinetic model of chemical and photochemical reactions, based on published experimental and *ab initio* data, for the UVC synthesis of adenine from HCN in water solvent within a lipid vesicle permeable to HCN, H${}_{2}$O and formimidic acid (the photon-tautomerized hydrolysis product of HCN), but impermeable to the reaction products, floating at the surface of a hot ocean and under a continuous UVC flux. All physical conditions chosen were consistent with those offered by the geochemical fossil evidence available from the early Archean.

The results presented here indicate that given UVC light continuously incident over a dilute aqueous solution of HCN at high temperature, significant dissipative structuring of adenine will occur, and if this occurs within a lipid vesicle enclosure, significant concentrations of adenine will build up within a short time period. There is no need to begin with large initial concentrations of HCN by invoking low temperature eutectic concentration and there is no need for alkaline conditions to favor HCN polymerization over hydrolysis since successive hydrolysis leads to formamide and ammonium formate which are catalysts for the important final step of the attachment of the last HCN to AICN (reaction # 13) to form adenine at high temperature. Destruction of adenine through hydrolysis does not compete significantly with its production through this proposed route, and, in fact, provides a route to the synthesis of guanine and certain amino acids.

Perturbations caused by the vesicle floating into patches of higher concentration of HCN and formimidic acid that could have existed at isolated regions of the Archean ocean surface microlayer could have provoked the non-linear autocatalytic system into new states of higher adenine productivity. This leads to a discontinuous increase in “metabolism” of precursor HCN molecules from the environment transformed inside the vesicle into UVC pigment molecules. Evolution is also towards concentration profiles of product molecules with an absorption maximum closer to the peak intensity of the incident UVC spectrum and towards peaked conical intersections to internal conversion, both increasing the overall efficacy of dissipation of the incident UVC solar spectrum. The Glansdorff–Prigogine criterion mandating decreasing contributions to the entropy production due to the reorganization of the free forces (the chemical affinities over the temperature) is observed and determines local stability (Appendix A). The concentration of adenine within our vesicle increased by 5 orders of magnitude, from $10^{-10}$ to $10^{-5}$ M, over the very short period of only 30 Archean days and the *total* entropy production, including the most important contribution due to the flow of energy (photons) through the system, increased by more than 5 orders of magnitude over the same period (Figure 18).

For very low diffusion rates, there can be significant coupling of reactions with diffusion, leading to non-homogeneous distributions of some of the intermediate products, with greater concentration of these at the center of the vesicle. Such spatial symmetry breaking is another form of dissipative structuring and could facilitate yet further structuring such as polymerization of the nucleobases into nucleic acid.

Dissipative structuring under light, as the fundamental creative force in biology, appears to have been ongoing, from the initial dissipation at the UVC wavelengths of the Archean by the fundamental molecules of life, to the dissipation of wavelengths up to the red edge (700 nm) by the organic pigments of today [10,11,12]. Beyond the red edge, starting at about 1200 nm, water in the ocean surface microlayer absorbs strongly and dissipates photons into heat efficiently. There is, therefore, still a wavelength region between 700 and 1200 nm which remains to be conquered by future evolution of pigments. The simultaneous coupling of biotic with abiotic irreversible processes, such as the water cycle and ocean and air currents, culminating in an efficient global dissipating system known as the biosphere, increases further the efficacy of solar photon dissipation into the far infrared much beyond 1200 nm [10,25].

Empirical evidence for selection in nature towards states of increased dissipation exists on vastly different size and time scales. For example, the increase in photon absorption and dissipation efficacy of a plant leaf over its life-cycle [133], the proliferation of photon absorbing pigments over the entire surface of Earth, the correlation between ecosystem succession and increased dissipation [134,135], and the general increase of biosphere efficacy in photon dissipation over evolutionary history, including, for example, the plant-induced increases in the water cycle [26,136] and animal dispersal of nutrients required for pigment synthesis [13]. There is also evidence for this at the microscopic scale, for example in the increased rates of energy dissipation per unit biomass of the living cell over its evolutionary history [137]. Here I have suggested how evolutionary increases in dissipation occur even at the nanoscale, i.e., the sequential increase in photon dissipation at each step along the dissipative synthesis of the nucleobases from precursor molecules under a UVC photon potential.

Any planet around any star giving off light in the long wavelength UVC region, but with protection against shorter wavelength light which could destroy carbon-based molecules through successive ionization, should therefore have its own concentration profile of dissipatively structured carbon-based fundamental molecules (UVC pigments) whose characteristics would depend on the exact nature of the local UV environment and the precursor and solvent molecules available. Examples may include the sulfur containing UV pigments found in the clouds of Venus [138], the UV absorbing thiophenes [139] and the red chlorophyll-like pigments [140] found on the surface of Mars, the UVC and UVB absorbing poly-aromatic hydrocarbons (PAHs) found in the atmosphere and on the surface of Titan [141], on the surface of asteroids, and in interstellar space [12]. The observation that thiophenes and PAHs found on mars, on asteroids, and in space are of generally large size can be understood from within this non-equilibrium thermodynamic perspective since, without the possibility of vibrational dissipation through hydrogen bonding to solvent molecules, these molecules would have “grown” to large sizes through dissipative selection in order to support many low frequency vibrational modes which would increase dissipation by pushing the emitted photon energy towards the infrared.

Dissipative structuring, dissipative proliferation, and dissipative selection are the necessary and sufficient ingredients for an explanation in physical-chemical terms of the synthesis, proliferation, and evolution of organic molecules on planets, comets, asteroids, and interstellar space [12], and, in particular, for contributing to an understanding of the origin and evolution of life on Earth.

## Figures and Tables

**Figure 1 entropy-23-00217-f001:**
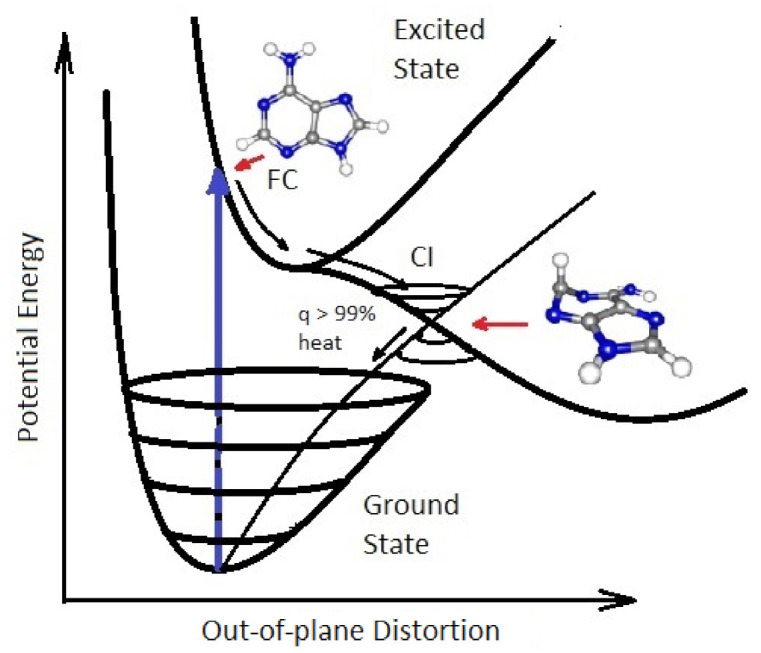
Conical Intersection (CI) for adenine showing the degeneracy of the electronic excited state with the electronic ground state after a UVC photon absorption event (blue arrow) which induces a nuclear coordinate deformation from its original structure in the Franck-Condon (FC) region, known as *pyramidalization*. Conical intersections provide rapid (sub-picosecond) dissipation of the electronic excitation energy into heat. The quantum efficiency (q) for this dissipative route is very large for the fundamental molecules of life, making them photochemically stable but more importantly very efficient at UV photon dissipation. Another common form of coordinate transformation associated with conical intersections are proton transfers within the molecule or with the solvent. Based on data from Andrew Orr-Ewing [29] Roberts et al. [30], Kleinermanns et al. [31] and Barbatti et al. [32]

**Figure 2 entropy-23-00217-f002:**
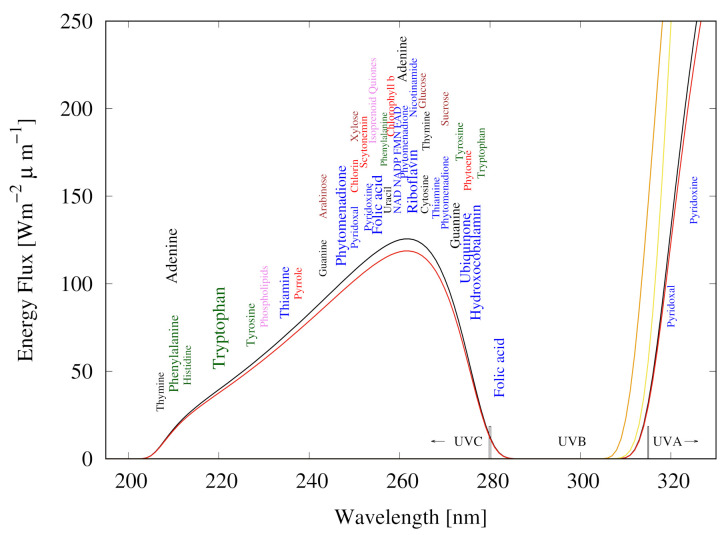
The spectrum of light available in the UV region at Earth’s surface before the origin of life at approximately 3.9 Ga and until at least 2.9 Ga (curves black and red respectively) during the Archean. CO${}_{2}$ and probably some H${}_{2}$S were responsible for absorption at wavelengths shorter than ∼205 nm and atmospheric aldehydes (common photochemical products of CO${}_{2}$ and water) absorbed between about 285 and 310 nm [35], approximately corresponding to the UVB region. Around 2.2 Ga (yellow curve), UVC light at Earth’s surface was extinguished by oxygen and ozone resulting from organisms performing oxygenic photosynthesis. The green curve corresponds to the present surface spectrum. Energy fluxes are for the sun at the zenith. The names of the fundamental molecules of life are plotted at their wavelengths of maximum absorption; nucleic acids (black), amino acids (green), fatty acids (violet), sugars (brown), vitamins, co-enzymes and cofactors (blue), and pigments (red) (the font size of the letter roughly corresponds to the relative size of their molar extinction coefficient). Indications that dissipative structuring occurred at the origin of life are that the absorption wavelengths of these fundamental molecules coincide with the Archean UV surface spectrum and that many have a peaked conical intersection to internal conversion. Adapted from Michaelian and Simeonov [12].

**Figure 3 entropy-23-00217-f003:**
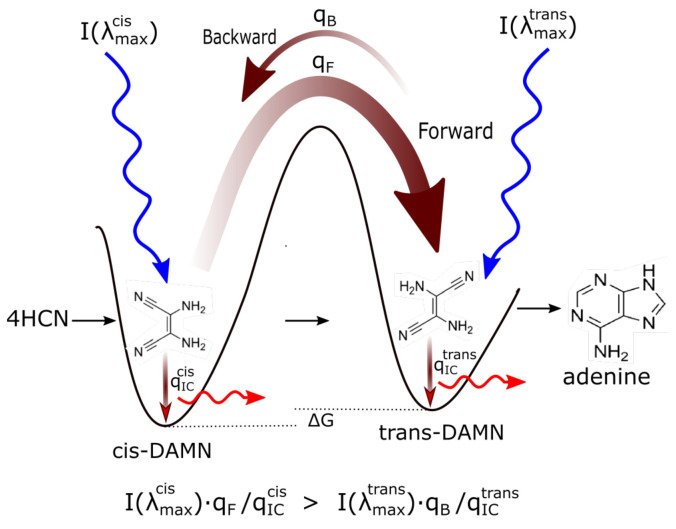
Mechanism for the evolution of molecular structures towards ever greater photon dissipative efficacy (microscopic dissipative structuring) on route to the fundamental molecules (in this case adenine, see Figure 4). The high activation barriers between configurations mean that reactions will not proceed spontaneously but only through coupling to photon absorption events. Forward and backward rates depend on photon intensities at the different wavelengths of maximum absorption I$\left(\lambda_{max}\right)$ for the two structures, and on the phase-space widths of paths on their excited potential energy surface leading to the conical intersection giving rise to the particular transformation, implying different quantum efficiencies for the forward (q${}_{F}$) and backward (q${}_{B}$) reactions. Since the intensity of the incident spectrum is assumed constant, and since $q_{F}+\cdots q_{IC}^{cis}=1$ and $q_{B}+\cdots q_{IC}^{trans}=1$ (where the “⋯” represents quantum efficiencies for other possible molecular transformations), those configurations (and also macroscopic concentration profiles) with greater photon dissipative efficacy (higher quantum efficiency for internal conversion q${}_{IC}$) will therefore gradually become more predominant under a continuously impressed UVC photon flux, independently of the sign or size of the difference in the Gibb’s free energies ΔG of the molecules.

**Figure 5 entropy-23-00217-f005:**
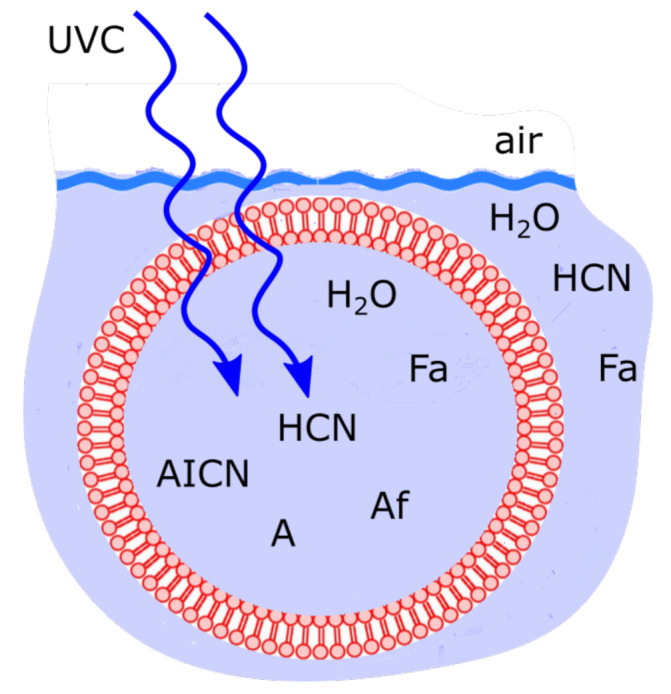
Fatty acid vesicle floating at the ocean surface microlayer, transparent to UVC light and permeable to H${}_{2}$O, HCN and formimidic acid (Fa) but impermeable to the photochemical reaction products (e.g., ammonium formate (Af), AICN, adenine (A)) which are larger in size and have larger dipole moments (Table 1).

**Figure 7 entropy-23-00217-f007:**
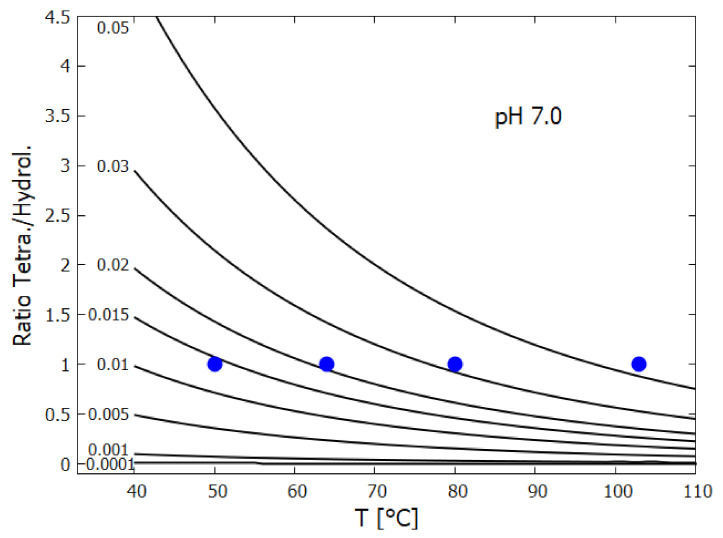
The ratio of the rates of tetramization to hydrolysis as a function of temperature as determined by our model for aqueous solutions of HCN at different concentrations [M] (given at the beginning of the corresponding trace) at pH 7.0. The experimental data points in blue for [HCN] = 0.05, 0.03, 0.02, 0.015 M at Ratio = 1 were obtained by linearly extrapolating to pH 7.0 from the closest two data points of Figure 15 of Sanchez et al. [68].

**Figure 8 entropy-23-00217-f008:**
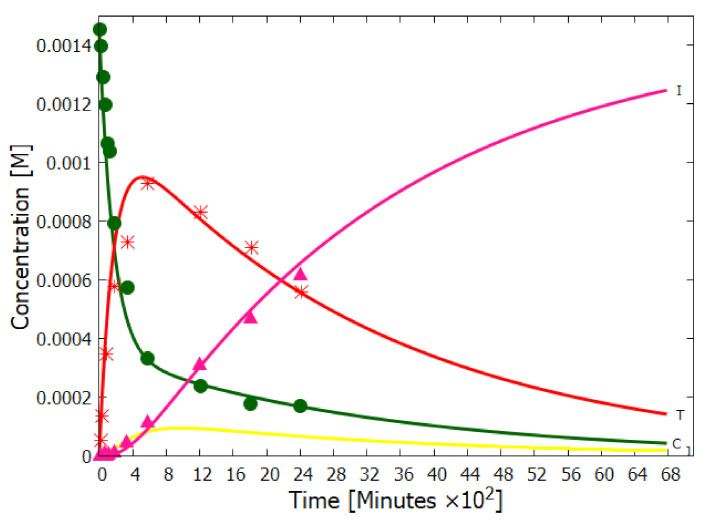
The concentrations of cis-DAMN (C, dark green), trans-DAMN (T, red), AIAC (J, yellow), and AICN (I, dark pink) obtained as a function of time from our model and compared with the experimental data points of Koch and Rodehorst (Figure 1 of reference [92]) starting with a concentration of cis-DAMN of 0.00145 M. The quantum efficiencies $q_{9r}$ and $q_{10}$ were adjusted to give the best fit. The overall light intensity was adjusted to give the correct time scale.

**Figure 9 entropy-23-00217-f009:**
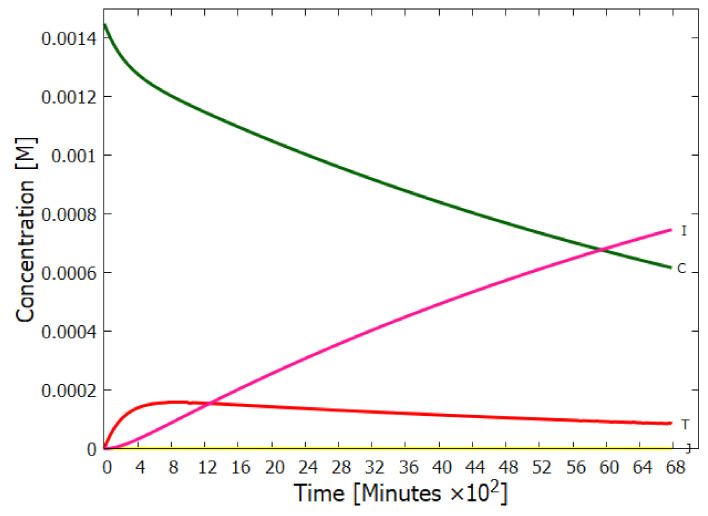
The concentrations of cis-DAMN (C, dark green), trans-DAMN (T, red), AIAC (J, yellow), and AICN (I, dark pink) obtained as a function of time from our model using the light spectrum of the Archean surface (Figure 2) starting with a concentration of cis-DAMN of 0.00145 M.

**Figure 10 entropy-23-00217-f010:**
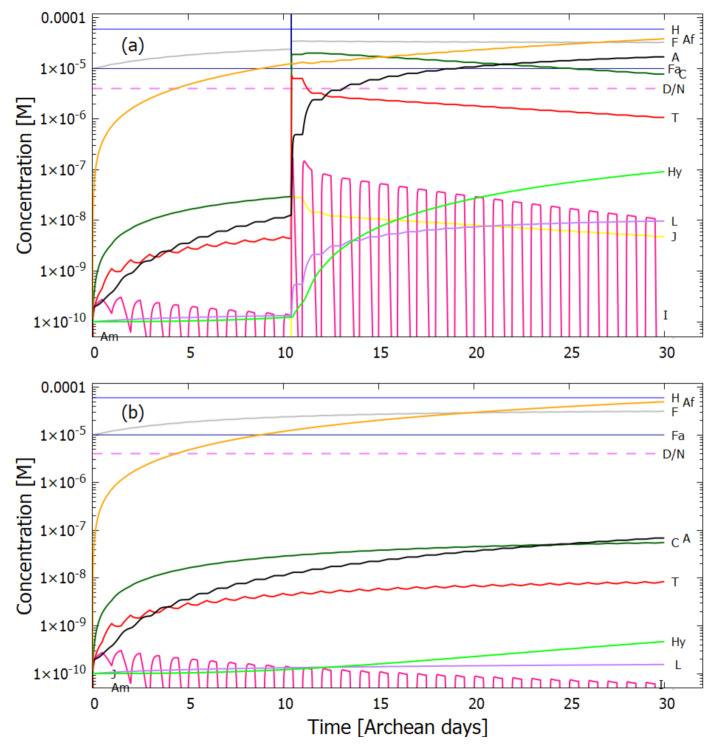
(**a**) Concentrations as a function of time in Archean days (16 h) of the precursor and product molecules; HCN (H—blue), formamide (F—gray), formimidic acid (Fa—dark blue), ammonium formate (Af—orange), cis-DAMN (C—dark green), adenine (A—black), AICN (I—dark pink), trans-DAMN (T—red), AIAC (J—yellow), AICA (L—purple), Amidine (Am—dark red), hypoxanthine (Hy—green), dissipatively structured on route to the synthesis of adenine (black trace). The initial conditions are temperature T=90 ${}^{\circ }$C, initial concentrations [H]${}_{0}=6.0\times 10^{-5}$ M, [F]${}_{0}=1.0\times 10^{-5}$ M, [Fa]${}_{0}=1.0\times 10^{-5}$ M and all other initial concentrations [Y]${}_{0}=1.0\times 10^{-10}$ M. The diffusion constant exponential factor was $1.0\times 10^{-6}$ (e.g., $D_{A}=1.0\times 10^{-6}$ cm${}^{2}$ s${}^{-1}$). There is one perturbation of the system corresponding to the vesicle floating into a region of HCN (H) and formimidic acid (Fa) of concentration 0.1 M for two minutes at 10.4 Archean days (vertical blue line at the top of the graph). A new stationary state at higher adenine concentration is reached after the perturbation. The violet horizontal dashed line, D/N, identifies alternate periods of daylight (violet) and night (blank). After 30 Archean days, the concentration of adenine within the vesicle (black trace) has grown by more than five orders of magnitude, from $1.0\times 10^{-10}$ to $1.7\times 10^{-5}$ M. (**b**) Same as (**a**) but without perturbation, giving a two orders of magnitude smaller final concentration of adenine compared to the case with perturbation (**a**).

**Figure 11 entropy-23-00217-f011:**
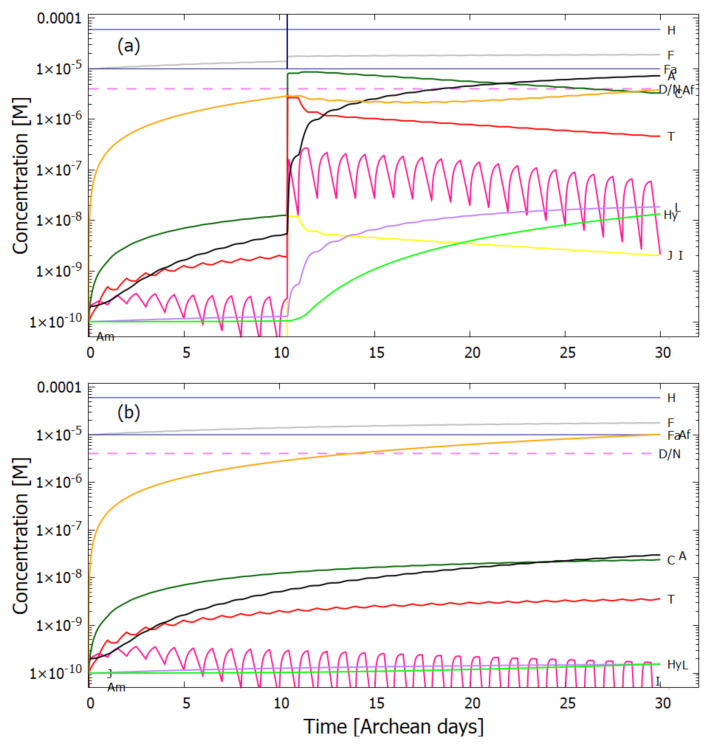
(**a**)The same as for Figure 10 except for a temperature of 80 ${}^{\circ }$C. The adenine concentration reaches $7.3\times 10^{-6}$ M. (**b**) The same without perturbation. The adenine concentration reaches $3.0\times 10^{-8}$ M. HCN (H—blue), formamide (F—gray), formimidic acid (Fa—dark blue), ammonium formate (Af—orange), cis-DAMN (C—dark green), adenine (A—black), AICN (I—dark pink), trans-DAMN (T—red), AIAC (J—yellow), AICA (L—purple), Amidine (Am—dark red), hypoxanthine (Hy—green).

**Figure 12 entropy-23-00217-f012:**
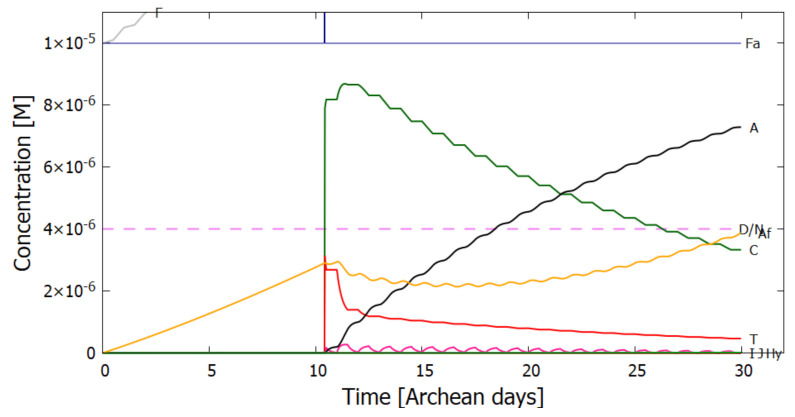
The same as for Figure 11a except plotted on a linear scale. There is a large increase in the rate of production of adenine (slope of black line) after the transient perturbation at 10.4 Archean days of 2 min duration. After the perturbation there is a greater metabolism of HCN (H) from the environment due to the non-linearity of the system. HCN (H—blue), formamide (F—gray), formimidic acid (Fa—dark blue), ammonium formate (Af—orange), cis-DAMN (C—dark green), adenine (A—black), AICN (I—dark pink), trans-DAMN (T—red), AIAC (J—yellow), AICA (L—purple), Amidine (Am—dark red), hypoxanthine (Hy—green).

**Figure 13 entropy-23-00217-f013:**
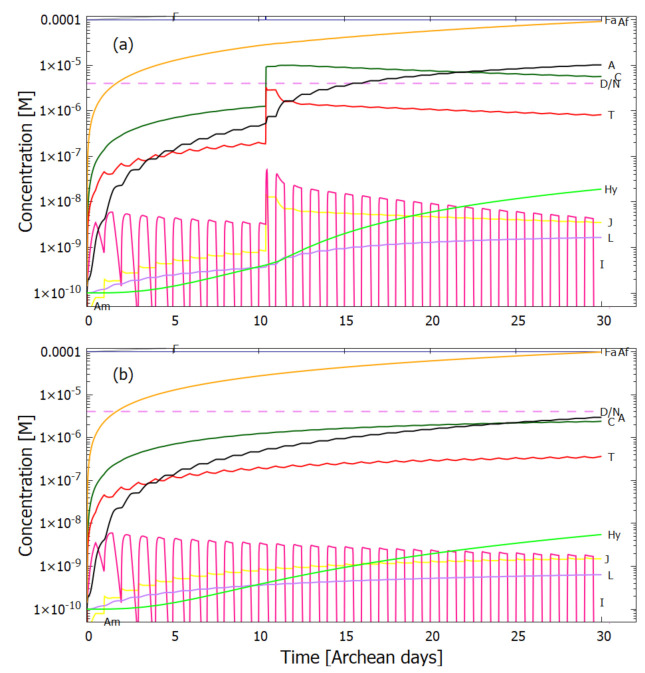
(**a**) The same as for Figure 11a, 80 ${}^{\circ }$C, except with concentrations of HCN (H) at $6\times 10^{-4}$ and formamide (F) and formimidic acid (Fa) at $1\times 10^{-4}$ M. After 30 days, the concentration of adenine (black trace) reaches a value of $1.0\times 10^{-5}$ M. (**b**) The same as (**a**) but without perturbation. The adenine concentration reaches $2.9\times 10^{-6}$ M. HCN (H—blue), formamide (F—gray), formimidic acid (Fa—dark blue), ammonium formate (Af—orange), cis-DAMN (C—dark green), adenine (A—black), AICN (I—dark pink), trans-DAMN (T—red), AIAC (J—yellow), AICA (L—purple), Amidine (Am—dark red), hypoxanthine (Hy—green).

**Figure 14 entropy-23-00217-f014:**
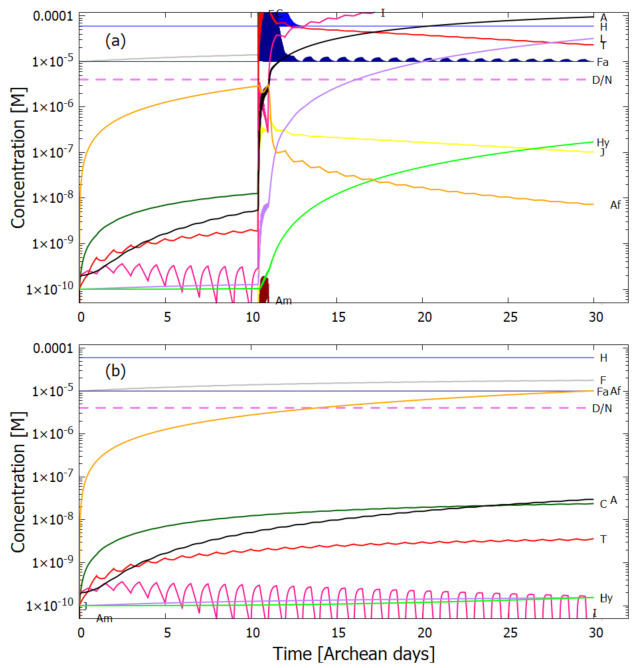
The same as Figure 11, 80 ${}^{\circ }$C, except with the diffusion exponential four orders of magnitude smaller, $1.0\times 10^{-10}$ cm${}^{2}$ s${}^{-1}$ (e.g., $D_{A}=1.0\times 10^{-10}$ cm${}^{2}$ s${}^{-1}$). Eleven bins in depth *x* below the ocean surface are plotted until reaching the bottom of the 100 μm (0.01 cm) vesicle. The top of the vesicle is at a depth of 0.00025 cm below the ocean surface. The small diffusion constant allows the observation of spatial symmetry breaking of the concentration profiles. This results in thicker lines since the 11 different depth bins are plotted in this figure. HCN (H—blue), formamide (F—gray), formimidic acid (Fa—dark blue), ammonium formate (Af—orange), cis-DAMN (C—dark green), adenine (A—black), AICN (I—dark pink), trans-DAMN (T—red), AIAC (J—yellow), AICA (L—purple), Amidine (Am—dark red), hypoxanthine (Hy—green).

**Figure 15 entropy-23-00217-f015:**
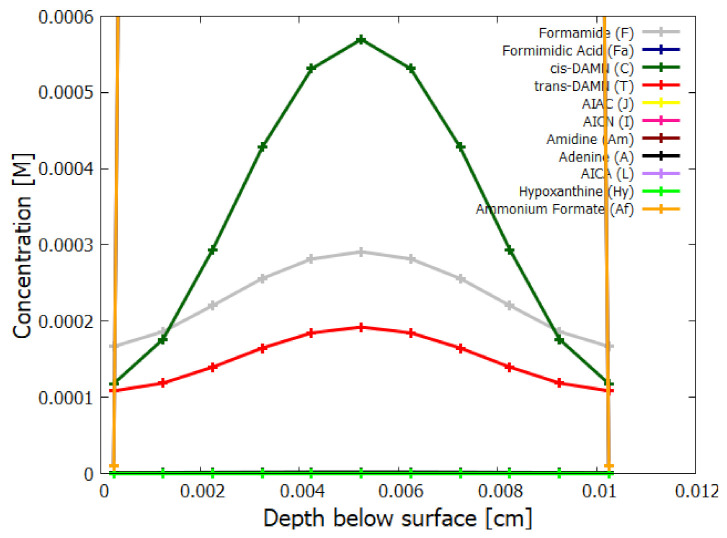
The concentration profile of the products as a function of depth below the ocean surface (the top of the vesicle is at a depth of 0.00025 cm below the surface) for the initial conditions of Figure 14 and taken at the time of 10.7 Archean days (5 h after the perturbation). Eleven bins in depth *x* below the ocean surface are plotted until reaching the bottom of the 100 μm (0.01 cm) diameter vesicle.

**Figure 16 entropy-23-00217-f016:**
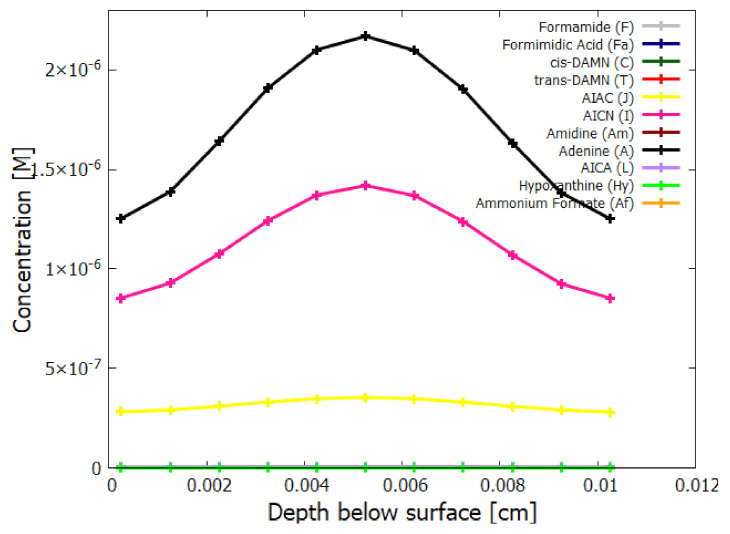
The same as Figure 15 except with an expanded y-scale to emphasize the products of lesser concentration taken at the time of 10.7 Archean days (5 h after the perturbation).

**Figure 17 entropy-23-00217-f017:**
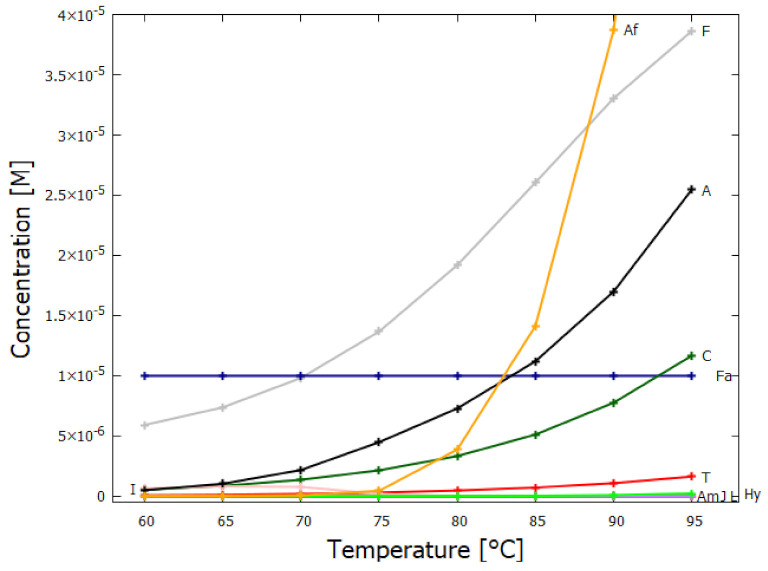
The temperature dependence of the concentrations of the product molecules obtained after 30 Archean days, with the initial conditions, [H]${}_{0}=6\times 10^{-5}$, [F]${}_{0}=1\times 10^{-5}$, [Fa]${}_{0}=1\times 10^{-5}$ M, and all other molecules [Y]${}_{0}=1\times 10^{-10}$ M and the diffusion exponential $1.0\times 10^{-6}$ cm${}^{2}$ s${}^{-1}$ (e.g., $D_{A}=1.0\times 10^{-6}$ cm${}^{2}$ s${}^{-1}$), with one perturbation at 10.4 Archean days. HCN (H—blue), formamide (F—gray), formimidic acid (Fa—dark blue), ammonium formate (Af—orange), cis-DAMN (C—dark green), adenine (A—black), AICN (I—dark pink), trans-DAMN (T—red), AIAC (J—yellow), AICA (L—purple), Amidine (Am—dark red), hypoxanthine (Hy—green).

**Figure 18 entropy-23-00217-f018:**
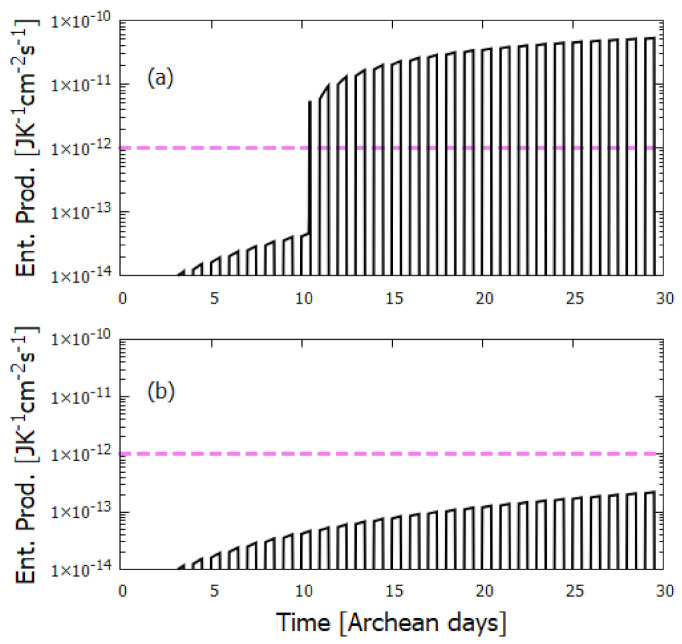
(**a**) The entropy production as a function of time during the UVC photochemical dissipative structuring process leading to adenine within a vesicle floating at the ocean surface at a temperature of 80 ${}^{\circ }$C. This entropy production increases monotonically as photochemical reactions convert HCN into the photon dissipative product molecules, including adenine. During the day, entropy production is due to the dissipation of the UV light into heat by the product concentration profile. At night, entropy production goes to zero (although thermal chemical reactions still occur during the night, this entropy production is small and not included). At 10.4 Archean days, the system is perturbed for two minutes and the entropy production increases discretely by almost 3 orders of magnitude and remains high. (**b**) The same but for no perturbation of the system.

**Table 2 entropy-23-00217-t002:** Reactions involved in the photochemical synthesis of adenine (see Figure 4). Temperature *T* is in K, all kinetic parameters *k* were obtained at pH 7.0, and *q* are the quantum efficiencies. (See descriptions of the derivations after table.)

#	Reaction	Reaction Constants
1	H $\overset{k_{1}}{⇀}$ F	$k_{1}=\exp\left(-14,039.0/T+24.732\right)$; s${}^{-1}$; hydrolysis of HCN [68,99,100]
2	$\gamma_{220}+$ F → Fa	$q_{2}=0.05$ [89,90,101,102,103]
3	$\gamma_{220}+$ Fa → H + H${}_{2}$O	$q_{3}=0.03$ [102,103,104]
4	F $\overset{k_{4}}{⇀}$ Af	$k_{4}=\exp\left(-13,587.0/T+23.735\right)$; s${}^{-1}$; hydrolysis of formamide [100,103]
5	4H $\overset{k_{5}}{⇀}$ C	$k_{5}=1/\left(\exp\left(-\Delta E/RT\right)+1\right)\cdot \exp\left(-10,822.37/T+19.049\right)$; M${}^{-1}$ s${}^{-1}$; ΔE=0.61 kcal mol${}^{-1}$ [68]
6	4H $\overset{k_{6}}{⇀}$ T	$k_{6}=1/\left(\exp\left(+\Delta E/RT\right)+1\right)\cdot \exp\left(-10,822.37/T+19.049\right)$; M${}^{-1}$ s${}^{-1}$; tetramization [68]
7	4H + T $\overset{k_{7}}{⇀}$ C+T	$k_{7}=\left(1.0/\left(1.0\times 0.01\right)\right)\exp\left(-\left(10,822.37-728.45\right)/T+19.049\right)$; M${}^{-2}$ s${}^{-1}$ [68]
8	4H + T $\overset{k_{8}}{⇀}$ 2T	$k_{8}=k_{7}$; M${}^{-2}$ s${}^{-1}$ [68]
9a	$\gamma_{298}$ + C → T	$q_{9}=0.045$ [92]
9b	$\gamma_{313}$ + T → C	$q_{9r}=0.020$ [68,71,92]
10	$\gamma_{313}$ + T → J	$q_{10}=0.006$ [68,71,92]
11	$\gamma_{275}$ + J → I	$q_{11}=0.583$; T →I; $q_{10}\times q_{11}=0.0034$ [68,71]
12	I $\overset{k_{12}}{⇀}$ L	$k_{12}=\exp\left(-E_{a}/RT+12.974\right)$; s${}^{-1}$; $E_{a}=19.93$ kcal mol${}^{-1}$; hydrolysis of AICN [69]
13	I:F + Af $\overset{k_{13}}{⇀}$ A + F	$k_{13}=\exp\left(-E_{a}/RT+12.973\right)$; M${}^{-1}$ s${}^{-1}$; $E_{a}=6.68$ kcal mol${}^{-1}$ [105,106]
14	I:F + Fa $\overset{k_{14}}{⇀}$ Am + Fa +H${}_{2}$O	$k_{14}=\exp\left(-E_{a}/RT+12.613\right)$; M${}^{-1}$ s${}^{-1}$; $E_{a}=19.90$ kcal mol${}^{-1}$ [107]
15	$\gamma_{250}$ + Am → A	$q_{15}=0.060$ [95]
16	A $\overset{k_{16}}{⇀}$ Hy	$k_{16}=10^{\left(-5902/T+8.15\right)}$; s${}^{-1}$; valid for pH within 5 to 8; hydrolysis of adenine [108,109]
17	$\gamma_{298}$ + C → C	$q_{17}=0.955$
18	$\gamma_{313}$ + T → T	$q_{18}=0.972$
19	$\gamma_{275}$ + J → J	$q_{19}=0.417$
20	$\gamma_{250}$ + Am → Am	$q_{20}=0.940$
21	$\gamma_{250}$ + I → I	$q_{21}=1.000$
22	$\gamma_{266}$ + L → L	$q_{22}=1.000$
23	$\gamma_{260}$ + A → A	$q_{23}=1.000$
24	$\gamma_{250}$ + Hy → Hy	$q_{24}=1.000$

**Table 3 entropy-23-00217-t003:** Diffusion constants relative to that of adenine for the different intermediate product molecules obtained from Equation (Equation 17). Two different multiplicative factors of $D_{A}=1\times 10^{-6}$ and $1\times 10^{-10}$ cm${}^{2}$ s${}^{-1}$ are used in the simulations.

$D_{H}$	$D_{F}$	$D_{\mathrm{Fa}}$	$D_{\mathrm{Af}}$	$D_{C}$	$D_{T}$	$D_{J}$	$D_{I}$	$D_{L}$	$D_{\mathrm{Am}}$	$D_{A}$	$D_{\mathrm{Hy}}$
7.752	2.988	11.190	6.689	0.892	4.073	4.073	1.908	1.532	1.000	1.000	2.482

**Table 4 entropy-23-00217-t004:** The concentration of adenine [M] produced in the vesicle after 30 Archean days at 80 ${}^{\circ }$C determined by the model as a function of a ±30% variation of the most sensitive parameters of the model with respect to their nominal values listed in the table (see also Table 2). The initial concentrations were [H]${}_{0}=6.0\times 10^{-5}$ M, [F]${}_{0}=1.0\times 10^{-5}$ M, [Fa]${}_{0}=1.0\times 10^{-5}$ M with all other concentrations [Y]${}_{0}=1.0\times 10^{-10}$ M and the diffusion constant was $D_{A}=1\times 10^{-6}$ cm${}^{2}$ s${}^{-1}$. One perturbation of the system of [H] and [Fa] to 0.1 M for 2 min occurs at 10.4 Archean days (see Figure 11).

#	Reaction	Parameter	Nominal Value	−30%	Nominal	+30%
9b	$\gamma_{313}$+T→ C	$q_{9r}$	0.020	8.222$\times 10^{-6}$	7.292$\times 10^{-6}$	6.528$\times 10^{-6}$
12	I $\overset{k_{12}}{⇀}$ L hydrolysis of AICN	$E_{a12}$	19.93 kcal mol${}^{-1}$	1.093$\times 10^{-6}$	7.292$\times 10^{-6}$	7.311$\times 10^{-6}$
13	I:F + Af $\overset{k_{13}}{⇀}$ A + F	$E_{a13}$	6.68 kcal mol${}^{-1}$	7.311$\times 10^{-6}$	7.292$\times 10^{-6}$	6.636$\times 10^{-6}$
14	I:F + Fa $\overset{k_{14}}{⇀}$ Am + Fa +H${}_{2}$O	$E_{a14}$	19.90 kcal mol${}^{-1}$	7.292$\times 10^{-6}$	7.292$\times 10^{-6}$	7.292$\times 10^{-6}$
15	$\gamma_{250}$+Am→ A	$q_{15}$	0.060	7.292$\times 10^{-6}$	7.292$\times 10^{-6}$	7.292$\times 10^{-6}$
16	A→ Hy hydrolysis of adenine	expn. of $k_{16}$	−5902	1.586$\times 10^{-9}$	7.292$\times 10^{-6}$	7.306$\times 10^{-6}$

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
