# Peer review of "The Dissipative Photochemical Origin of Life: UVC Abiogenesis of Adenine"

_entropy, 2021, doi:10.3390/e23020217_

Round 1
Reviewer 1 Report
My concerns with this MS remain. As indicated in my earlier report this work repeats much of the author's earlier work in support of the Prigogine dissipative non-equilibrium model for the emergence of life. To the extent that there is new material presented in this paper, it is hard to assess it, given the review like nature of the article and its length (52 pages), much of it repeating earlier ideas. Given the fact that Prigogine's contribution to the question is now close to 50 years old, for this work to be a useful contribution, it needs to be shortened significantly and any new elements emphasized as such. As I mentioned in my earlier review, reference to recent work which supports the alternative kinetic model for abiogenesis should be mentioned but it hasn't been. I indicated that the recent Moberg article in Angewandte Chemie commemorating 75 years to Schrodinger's What is Life book, summarizes that work lucidly and provides much of the relevant new literature on the topic (DOI: 10.1002/anie.201911112).
Without significant revision along the lines indicated, I cannot recommend the MS for publication.
Author Response
I thank the reviewer for their review and for their suggestions which have helped improved the manuscript.
I have included a new paragraph in the Introduction beginning on line 35 giving a citation to the review article by Moberg and included citations to other works which have been concerned with the application of non-equilibrium thermodynamic theory to living systems. As mentioned in my previous reply to the reviewer, within the non-equilibrium thermodynamic framework, it is known that kinetic factors like autocatalysis become more important than thermodynamic improbability and this can be ultimately related to increases in dissipation. This was included in a new paragraph beginning on line 287. The kinetic model and Classical Irreversible Thermodynamic theory are therefore not in as much contradiction as might initially seem.
I have included a subtitle and rewritten the Abstract and various parts of the main text (e.g. line 45, line 142, line 996, and line 1002) to emphasize explicitly what is new in the manuscript. Concerning the manuscript length and including Prigogine theory, the manuscript is meant to be a contribution to the special issue of Entropy that I am editing entitled “Dissipative Structuring in Life”. Although written in first instance for an audience of the readers of Entropy, some of whom would have knowledge of non-equilibrium thermodynamics, I believe that a brief introduction to the most important aspects of this formalism, particularly concerning the mechanism of evolution in these non-linear out-of-equilibrium systems, seems appropriate for those readers who may not have knowledge. Also, in the analysis of the results of the synthesis of adenine, I frequently refer back to specific equations presented in the thermodynamic section, so its inclusion is important for a complete understanding of the results. However, I have, on the suggestion of the reviewer, gone through the manuscript and removed non-essential parts and redundancies, and re-written other parts making them more concise. The manuscript has now been reduced to 46 pages from initially 52, albeit some of that difference has to do with reformatting some of the figures.
I once again thank the reviewer for their revision and consideration.
Reviewer 2 Report
The results have been presented in the article, indicating that given UVC light continuously incident over a dilute aqueous solution of HCN at high temperature, significant dissipative structuring of adenine will occur, and if this occurs within a lipid vesicle enclosure, significant concentrations of adenine will build up within a short time period. Furthermore, destruction of adenine through hydrolysis and effects of perturbations caused by the vesicle have been reported in the article. I am interested in the UV photochemical dissipative structuring, proliferation, and evolution of molecules on route to the nucleobase adenine from the common precursor molecules HCN and H2O occurring within a fatty acid vesicle. Therefore, I consider that this study could make a significant contribution to the origin of life. However, in the beginning, I could not well understand the content of the article, probably because both the title and the abstract are largely deviated from the content, especially of the results.
So, I would like to recommend for the author to revise the title and the abstract, if the author can agree with my suggestions.
Major comments:
- I recommend to change the title of the manuscript or at least to add a subtitle to coincide with the content.
- The abstract should be also revised to coincide with the content, for example, as referring the sentences after line 972, “As an example, I presented a simple kinetic model of UVC photochemical reactions,”of “ Summary and conclusions”.
- I would like also to recommend the author to consider about my thoughts described below to rewrite two paragraphsin Summary and conclusions, if the author could agree with my idea and if possible, becauseit seems to me that the author’s idea about the origin of life is somewhat myopic, although it is unnecessary to revise the manuscript.
From line 1045: We have considered the synthesis of such a basis set of dissipative molecules to be the first step of incipient life and, therefore, under this criterion, incipient life has already been discovered on the other bodies of our solar system, for example, in the sulfur containing UV pigments found in the clouds of Venus etc.
The author also describes from line 1058 that dissipative structuring, dissipative proliferation, and dissipative selection are the necessary and sufficient ingredients for an explanation in physical-chemical terms of the synthesis, proliferation, and evolution of organic molecules on planets, comets, asteroids, and interstellar space, and, in particular, for explaining the origin and evolution of life on Earth.
The reason is because I consider that, although dissipative structuring, dissipative proliferation, and dissipative selection should be the necessary and sufficient ingredients for an explanation in physical-chemical terms of the synthesis, proliferation, and evolution of organic molecules as the author describes, those would not be the sufficient condition for explaining the origin and evolution of life on Earth, because it should be indispensable to explain at least the establishment process of the “Central dogma” composed of gene and protein in order to understand the origin of life, which cannot be always explained with dissipative structuring and so on.
Minor revisions:
- In Figure 1, I cannot observe the green curve, which isdescribed in the Figure legend.
- Page 9, line 304: figure 2 should be written in black letters.
- In Figure 7, which curve is forAIAC (J)is not shown.
Author Response
I thank the reviewer for their review of my manuscript, for their favorable evaluation of its importance, and for their useful suggestions for improvement.
Major Revisions
I have included a subtitle and re-written the abstract to better indicate the contents of my manuscript. I have also extensively rewritten the main text in order to emphasize better the new content and to make the text more concise. The text has been reduced to 46 pages from the original 52, which also includes better formatting of some figures.
The two paragraphs (beginning now on lines 1051 and 1065) in the Summary and Conclusions have been rewritten to accommodate the concerns of the reviewer. In particular, I have removed the sentence containing “…to be the first step of incipient life and, therefore, under this criterion, incipient life has already been discovered on the other bodies of our solar system ”. I have also removed reference to (line 1065) “… explaining the origin and evolution of life on Earth.” which instead I now write as “…contributing to an understanding of the origin and evolution of life on Earth.”
Minor Revisions
- I have corrected the green color of the curve of Figure 1
- On line 304 I have written “figure 2” in black letters.
- Figure 7 has been corrected to see better the AIAC label “J”, which is the yellow line. I have also now specified the colors of each component in the figure legends of both figures 7 and 8.
I thank the reviewer once again for their revision and consideration of my manuscript.
Reviewer 3 Report
A dissipative photochemical origin of life.
The manuscript by Karo Michaelian “A dissipative photochemical origin of life” deal with the tentative to answer to the essential questions “what is life?” and how the matter organization leading to “living matter” (abiogenesis) started. Some hypotheses concerning abiogenesis have formulated in time ranging from the arrival of molecules from the space to the purely accidental synthesis of molecules that have evolved towards increasingly complex aggregates with the unique property to transmit information. The author analyses a model system to support the structuring of fundamental early “life” molecules as a dissipative process. UV light, instead of to be considered only harmful, plays the role of driving force. The author is quite extreme on this and writes “Thus, rather than requiring refuge or protection from this UV light, it is argued here that UV-induced molecular transformations providing innovations which allowed early molecular life to maximize UV exposure…”. The manuscript is interesting, timely and poses questions and results that are certainly suggestive for further discussions and development. The model system proposed and analyzed is intriguing; the used hypothesis and ineluctable approximations are discussed in detail and the conclusions are very interesting. In my opinion, it should be published in Entropy after that some points will be clarified.
General comment.
I realize that this general comment is a matter of taste, but I think that the manuscript could be more readable and direct to the objective if some of the arguments treated in the main text become appendixes. As an example, in my opinion, part two and three could be shortened in the main text to stress main points and conclusions, transferring the detailed discussion, obviously useful for the reader and the development of the model, in one or two appendices. I underline that this comment is only a personal suggestion and is far from mandatory to accept the manuscript.
Line 54. “Boltzmann recognized … and suggested that…”. Citation [11] is not Boltzmann.
Line 113. If I understand correctly, the author contrasts the “…natural selection leading to better adapted…” with “… dissipative structuring of material under the imposed UV…”. Perhaps this is only an apparent contrast as “dissipative structuring of material” can be viewed as a sort of adaptation.
Line 321. “The final product in the photochemical synthesis … must, however, always, have…”. I am puzzled on this proposition because taking chlorophyll as an example, the molecule is not photo-stable, has an excited state lifetime of around 5 ns in a solution that shortens drastically when inserted in a chlorophyll-protein complex, due to the rapid transfer of the electronic excitation toward neighbour molecules, with carotenoids that act to protect chlorophyll against photo-damage. Isn't chlorophyll a fundamental molecule of life?
Line 419. The model uses vesicles enclosed by a double layer of fatty acids and having a diameter of around 100μm. This is the depth of the microlayer of the ocean surface: is the dimension of the vesicles a reasonable hypothesis? How the dimension of the vesicle has an impact, if any, on the dynamical results of the treated model? It is also assumed that this double layer of fatty acids is permeable to water (see also caption of figure 4 and line 974 and impermeable to the photochemical reaction product. However, water diffusion across phospholipids double layers is slow and the water exchange through the membrane is across specialized protein structure, the aquaporin.
Table 2. Please, insert in the table caption that q=quantum yield.
Line 515. ln(1/T)? Probably is the graph of 1/T vs ln(k).
Line 523. The choice of the Boltzmann factor defined as 1/(exp(∆E/RT)+1) is not clear to me.
Kinetic equations (between line 663 and line 664). In the description of the actors involved in the kinetic equations, the meaning of D should be indicated as well as that the first right term of all equation represents the diffusional contribution. In this model, the products of the set of reactions accumulate inside the vesicles as the bilayer is not permeable. This will presumably lead to a “dynamical” change in the viscosity of the inner medium (as is also written in line 682). If this is true, the choice of unique values for the diffusion constants could be a problem? Moreover, I presume that the H and Fa input is through the diffusion term acting with the boundary condition described between lines 700 and 704. However, during the time course of the reactions, both H and Fa should diffuse across the lipid double layer with a diffusion coefficient that will be different concerning that of the cytoplasm. This could be also a problem for the “concentration spikes” of H and Fa.
Line 762: It is not clear to me if the results in figure 7 have been obtained taking into account the subset of reactions from #9a to #11 only and the respective kinetic equations. Also, if the parameters are adjusted to give the best fit of the model to the experimental data “… agree well with experiment” is a necessity. But, perhaps, I am confused.
Figure 7- Figure 8. In my opinion, the figure is more readable if the different concentration of C, T, J, I are associated with colours in the figure caption. This is also for the other figures.
Line 809. All the preceding description/discussion is in terms of a temperature of 80°C but the first figure of the concentration dynamics obtained is for 90°C. Why?
Figure 9 vs figure 10. The time evolution of “I” show oscillations due to the day/night cycle with concentration fall below 10-10 during night period at 90°C. Instead, at 80°C this dynamic is different with relatively small oscillations. This is true also comparing figure 12.
Calculations with “sudden impulses” of HCN and Fa at a time different to that reported in figures have been performed?
Line 836-840: I agree with the conclusion written here that this model system show bifurcation after perturbation, but it is not immediately obvious why this new stationary state is “… more probable than the initial…”
Question. As the reaction products accumulate in the vesicles, it is possible to say that these products can reach an equilibrium state?
Author Response
I thank the reviewer for their thorough revision of my manuscript and for their suggestion that it is interesting and could be published in Entropy. I found it refreshing to be provided with such a constructive and careful review. I have taken into account the suggestions and corrections of the reviewer as described below.
Reviewer: I realize that this general comment is a matter of taste, but I think that the manuscript could be more readable and direct to the objective if some of the arguments treated in the main text become appendixes. As an example, in my opinion, part two and three could be shortened in the main text to stress main points and conclusions, transferring the detailed discussion, obviously useful for the reader and the development of the model, in one or two appendices. I underline that this comment is only a personal suggestion and is far from mandatory to accept the manuscript.
Reply: The detailed discussion of the non-equilibrium thermodynamics has now been transferred to an Appendix. Furthermore, I have reduced the length of the Introduction by removing the thermodynamic description and instead moving it to a new section (section 3) where I present the general framework employed, including only a brief summary of the non-equilibrium thermodynamics and a new figure (Fig. 3) which helps in the understanding the greater stability of dissipative systems . The old section 3 (now section 2) on the photochemistry was left in the main text since details from it, particularly conical intersections, are referred to frequently in most following sections.
Reviewer: Line 54. “Boltzmann recognized … and suggested that…”. Citation [11] is not Boltzmann
Reply: The original citation to Boltzmann was to an accessible English translation. I now include a second citation to a reproduction of the original reference in German (line 183).
Reviewer: Line 113. If I understand correctly, the author contrasts the “…natural selection leading to better adapted…” with “… dissipative structuring of material under the imposed UV…”. Perhaps this is only an apparent contrast as “dissipative structuring of material” can be viewed as a sort of adaptation.
Reply: The reviewer is correct in that, from the perspective of the dissipative structure (organism), this can be seen as a sort of adaption under the impressed UVC flux. However, it is not survival under the adverse conditions that is being selected (nature could do that y simply making the structure (organism) transparent or reflective to this UVC light), but rather transformation which improves photon dissipation. A new figure (Fig. 3) is included to explain this in simple terms and I have also included a new paragraph of explanation beginning on line 220 in the text.
Reviewer: Line 321. “The final product in the photochemical synthesis … must, however, always, have…”. I am puzzled on this proposition because taking chlorophyll as an example, the molecule is not photo-stable, has an excited state lifetime of around 5 ns in a solution that shortens drastically when inserted in a chlorophyll-protein complex, due to the rapid transfer of the electronic excitation toward neighbour molecules, with carotenoids that act to protect chlorophyll against photo-damage. Isn't chlorophyll a fundamental molecule of life?
Reply: It may have been that pyrrole was the original fundamental dissipative structure, having a broad pi->pi* singlet-singlet transition at 237.5 nm [1], with a conical intersection to ultra-rapid internal conversion [2]. It’s structuring into five-membered hetero-cycles, like protoporphyrin [3] and then chlorophyll b, probably came later. We have suggested that dissipation gradually progressed towards visible wavelengths after more complex biosynthetic pathways emerged since visible wavelengths have limited capacity to transform carbon based covalently bonded material. As part of this evolutionary process, we have suggested the formation of complexes consisting of antenna molecules (e.g. chlorophyll or amino acids) attached to acceptor molecules (e.g. carotenoids or DNA) which do have a conical intersection to internal conversion. We have published a paper in Entropy [4] suggesting this as a possible reason for the stereochemical era involving aromatic UVC absorbing amino acids with strong chemical affinity to their cognate codons.
[1] P. A. Mullen and M. K. Orloff, Ultraviolet Absorption Spectrum of Pyrrole Vapor Including the Observation of Low-Energy Transitions in the Far Ultraviolet, J. Chem. Phys., 51, 1969.
[2] Vallet et al., Photochemistry of pyrrole: Time-dependent quantum wave-packet description of the dynamics at the 1pi-sigma*-S0 conical intersections, J. Chem. Phys., 123, 144307, 2005.
[3] Lozovaya et al., Protoporphyrin IX as a possible ancient photosensitizer: spectral and photochemical studies, Origins of life and evolution of the biosphere, 20, 321-330, 1990.
[4] Mejía, J. and Michaelian, K. Photon Dissipation as the Origin of Information Encoding in RNA and DNA. Entropy, 2020, 22, 940.
Reviewer: Line 419. The model uses vesicles enclosed by a double layer of fatty acids and having a diameter of around 100μm. This is the depth of the microlayer of the ocean surface: is the dimension of the vesicles a reasonable hypothesis? How the dimension of the vesicle has an impact, if any, on the dynamical results of the treated model? It is also assumed that this double layer of fatty acids is permeable to water (see also caption of figure 4 and line 974 and impermeable to the photochemical reaction product. However, water diffusion across phospholipids double layers is slow and the water exchange through the membrane is across specialized protein structure, the aquaporin.
Reply: The size of the fatty acid vesicle was chosen to be large in order to conveniently observe spatial symmetry breaking in the concentration profiles as a function of vesicle radius after the initial spike in the concentrations of H and Fa for the smallest diffusion constant used, given the finite time step used in the program. Other than this, there is no impact of size on the dynamical concentration results. The 100 um size chosen is somewhat on the large size, however, routinely observed giant fatty acid vesicles of approximately 30 um are found. Szostak and coworkers [1] have observed filamentous growth of immobilized fatty acid vesicles with tails of the order of 100 um. We have now included a statement at line 574 indicating the scale invariance of our model.
Single tailed fatty acid membranes that are much more dynamic than double tailed phospholipid membranes because flip-flop rates in fatty acids are significantly greater. This leads to membranes that are much more dynamic than phospholipid membranes [1]. It would then be expected that the rate of diffusion H2O across a fatty acid vesicle would be greater than that for phospholipids. However, the model does not require the diffusion of water across the bi-layer, although it does require diffusion of HCN, which, in fact, has a larger dipole moment than that of water.
[1] Christian Hentrich and Jack W. Szostak, Controlled Growth of Filamentous Fatty Acid Vesicles under Flow, Langmuir. 30 (49) 14916–14925 (2014).
Reviewer: Table 2. Please, insert in the table caption that q=quantum yield.
Reply: The table caption has been modified to include the definition of q.
Reviewer: Line 515. ln(1/T)? Probably is the graph of 1/T vs ln(k).
Reply: This has been corrected (new line 421).
Reviewer: Line 523. The choice of the Boltzmann factor defined as 1/(exp(∆E/RT)+1) is not clear to me.
Reply: The ratio of the fraction of thermal tetramizations of trans-DAMN (t) to cis-DAMN (c) is;
F_t/F_c = exp^{-(E_t – E_c)/RT} = exp^{-Delta E/RT}
and since F_t + F_c = 1
then F_c*exp{-Delta E} + F_c = 1
therefore, F_c = 1/(exp^{-Delta E/RT +1), and similarly, F_t = 1/(exp^{+Delta E/RT +1).
Reviewer: Kinetic equations (between line 663 and line 664). In the description of the actors involved in the kinetic equations, the meaning of D should be indicated as well as that the first right term of all equation represents the diffusional contribution. In this model, the products of the set of reactions accumulate inside the vesicles as the bilayer is not permeable. This will presumably lead to a “dynamical” change in the viscosity of the inner medium (as is also written in line 682). If this is true, the choice of unique values for the diffusion constants could be a problem? Moreover, I presume that the H and Fa input is through the diffusion term acting with the boundary condition described between lines 700 and 704. However, during the time course of the reactions, both H and Fa should diffuse across the lipid double layer with a diffusion coefficient that will be different concerning that of the cytoplasm. This could be also a problem for the “concentration spikes” of H and Fa.
Reply: I have now described the meaning of D and also state that the first term of the reaction equations represents the diffusion flow (line 579). If the concentrations of the product molecules within the vesicle arrive at very large values, then the reviewer would be correct, the diffusion constants would become dynamic. For the reactions considered here, over the time period of 30 Archean days, concentrations of the large product molecules do not become greater than 10^{-5} M, which probably has little effect on diffusion. A more refined future model could consider dynamic diffusion and more individualized molecular membrane permeabilities. A statement to this effect is included at line 614.
Reviewer: Line 762: It is not clear to me if the results in figure 7 have been obtained taking into account the subset of reactions from #9a to #11 only and the respective kinetic equations. Also, if the parameters are adjusted to give the best fit of the model to the experimental data “… agree well with experiment” is a necessity. But, perhaps, I am confused.
Reply: In fact, the full model containing all reactions was used with the temperature set to 20°C (experiment) and all initial concentrations set to zero except cis-DAMN which was set to the initial value of experiment. The shape of the incident spectrum was adjusted to give that of the Rayonet lamp. This simulates the conditions of experiment with the full model. Reaction #12 essentially does not occur at this low temperature. Reactions #13 and #14 will not occur since there is no F or Af or Fa. The only free parameters used to fit the experimental data were the quantum efficiencies of photo-reaction #9b and #10 since these values were not available from the literature. The overall intensity of the light on sample was also adjusted since this could not be determined from the experimental paper. Changing the overall light intensity only expands or contracts the time-scale (the x-scale), but does not change the relative shapes of the curves. I have re-written the corresponding sentence on line 699 to read, “The results, plotted in figure 8 (old figure 7), indicate that our full model, employing the initial conditions of experiment, is able to reproduce well the shapes of the three experimental data sets by fitting only two parameters, the quantum efficiencies q_9r and q_10.”
Reviewer: Figure 7- Figure 8. In my opinion, the figure is more readable if the different concentration of C, T, J, I are associated with colours in the figure caption. This is also for the other figures.
Reply: I have now included the color code in all figure captions.
Reviewer: Line 809. All the preceding description/discussion is in terms of a temperature of 80°C but the first figure of the concentration dynamics obtained is for 90°C. Why?
Reply: There is significant uncertainty in supposing a date for the origin of life or the temperature of Earth’s surface at that time. There is a consensus, however, that it occurred after “the late lunar bombardment” at 3.9 Ga in either thermal or hyperthermal conditions, with Earth's surface cooling throughout the Archean. It would therefore be relevant to consider whether the dissipative structuring of adenine proposed here could have been efficient over a range of temperatures (60 - 95 °C), including, perhaps, conditions which may have existed even before the event of the origin of life. I have included a new paragraph stating this beginning at line 657 in the subsection “Initial Conditions”.
Reviewer: Figure 9 vs figure 10. The time evolution of “I” show oscillations due to the day/night cycle with concentration fall below 10-10 during night period at 90°C. Instead, at 80°C this dynamic is different with relatively small oscillations. This is true also comparing figure 12.
Reply: The concentrations are plotted on a logarithmic scale. Consideration of the concentrations of “I” (AICN) after the perturbation at 10.4 Archean days shows that at 90 °C the initial concentration is about 3 times smaller than that after the same perturbation at 80 °C. This is due to the fact that “I” is used up in reactions #12 to #14, the rates of which increase with increasing temperature. The logarithmic concentration y-scale, the different initial values after the perturbation, and the greater use of “I” during the night due to reactions #12 to #14 for the higher temperature 90 °C run leads to what appears (somewhat misleadingly due to scale) to be much larger oscillations in “I” at high temperature.
Reviewer: Calculations with “sudden impulses” of HCN and Fa at a time different to that reported in figures have been performed?
Reply: In an earlier version of this article I had included various smaller impulses at different times, each giving different but generally similar dynamics (e.g. different if the impulse occurred at night or day and depending on their size), however, this caused a lot of cluttering of the graphs, making them more confusing and a reviewer requested that I remove the clutter from the graphs, which I did by going to only one larger perturbation.
Reviewer: Line 836-840: I agree with the conclusion written here that this model system show bifurcation after perturbation, but it is not immediately obvious why this new stationary state is “… more probable than the initial…”
Reply: I have included a new diagram (figure 3) which explains why those concentration profiles which have a higher number of molecules with conical intersections to internal conversion would be more stable/probable than other configurations. Also, two new paragraphs are included beginning at lines 220 and 232.
Reviewer: Question. As the reaction products accumulate in the vesicles, it is possible to say that these products can reach an equilibrium state?
Reply: Rather than an equilibrium state, I would call it a non-equilibrium stationary state at which the product concentrations no longer increase over time. This would be caused by product concentrations within the vesicle building up to such an extent that diffusion from the vesicle into the surrounding water would occur (not included in the model) even for those products with large size and dipole moments. This could be imagined as vesicles seeding the ocean surface with dissipative pigments or fundamental molecules. Another possibility is that of product build-up in the interior causing the vesicle to divide into two. Such dynamics, however, would have to be considered in a more detailed future version of the model. I have added a paragraph to suggest this in the text at line 614.
Round 2
Reviewer 1 Report
As I mentioned in my earlier review, this paper would need to be drastically revised and significantly shortened to be a useful contribution. That has not been done so I cannot recommend acceptance
Author Response
As mentioned in my previous reply to the reviewer, there are no sections which can be removed from my manuscript without compromising the understanding it intends to portray, although I did make a sincere effort to reducie the length. As I also mentioned, my paper is intended to be the first contribution to a special edition of Entropy to be entitled “Dissipative Structuring in Life” and so should present those elements of Classical Irreversible Thermodynamic Theory necessary to understand the concept of dissipative structuring for the other papers to follow. I have however, now moved the detailed thermodynamic description to an appendix, reducing significantly the length of the main text. I would further like to point out that there are no size limits on articles acceptable to Entropy.
Reviewer 2 Report
The authors have extensively and appropriately revised the previous manuscript as accepting my comments. So, I would like to recommend the editor to accept the revised manuscript.
Author Response
I thank the reviewer for recommending acceptance of my manuscript.
Reviewer 3 Report
The author answered to all points raised and has modified his manuscript extensively taking into accout the suggestions and improving the readability. I think that is eligible to publication in Entropy.
This manuscript is a resubmission of an earlier submission. The following is a list of the peer review reports and author responses from that submission.
Round 1
Reviewer 1 Report
This article by Michaelian, proposing a dissipative photochemical origin of life, is a worthy study, and expresses quite clearly the extended knowledge of the author on the subject matter described. Aspects of the work have appeared in earlier publications, in particular refs. 7 and 8, but the importance of the topic, one that continues to challenge contemporary science, does justify further discussion. The essence of the author’s proposal is that the origin of life is a thermodynamic process, primarily describable by Prigogine’s non-equilibrium thermodynamics, and belongs to the ‘thermodynamic origin of life’ school of thought. Moreover, the author claims that natural selection and evolution also derive from thermodynamic considerations, not kinetic ones. I have a problem with that, though my saying so may just reflect my own personal view on this topic. Nonetheless I’d say that the MS would be significantly improved if a section explaining how the non-equilibrium approach to selection and evolution compares with the kinetic/Darwinian approach, pointing out, if possible, advantages of the thermodynamic approach. Ultimately, with two opposing scenarios for the driving force for life’s emergence, some attention to that deep divergence would be beneficial for supporters of either of the two approaches. A useful recent reference that succinctly reviews the life phenomenon, mentioning both kinetic and thermodynamic approaches, can be found in C. Moberg, Angew. Chemie, Int. Ed., 59, 2550-3, 2019, and references therein.
.
Author Response
I thank the reviewer for their review and remarks concerning my manuscript.
I would like to mention to the reviewer that they and I are probably in general agreement concerning the importance of kinetics to the origin and evolution of life. In isolated systems governed by equilibrium thermodynamics, the evolution of the system is determined only by thermodynamics and the result is independent of the initial conditions or kinetic factors. However, in out-of-equilibrium thermodynamics, particularly in the non-linear regime of Classical Irreversible Thermodynamics, kinetics plays a very important part in the evolution of the system. Kinetic factors, like auto-catalytic activity, can become more important than thermodynamic improbability. In this non-linear regime, there are multiple, locally stable, stationary states available and the system can evolve from one stationary state to another depending on the size of a fluctuation near a bifurcation and the kinetic factors, as is evident in Figures 6 through 12 in the present manuscript. This is now emphasized in a new paragraph, second to last at the end of the section “Thermodynamic Foundations” (line 253).
In other works I have compared the non-equilibrium thermodynamic approach with that of Darwin’s theory of natural selection [1,2]. Since including this comparison in detail would distract attention from the central theme and increase the size of the manuscript, I have instead included a brief and general comparison of the two approaches in the Introduction and Conclusions of the new version of the manuscript.
I thank the reviewer once again for their suggestions and remarks which have helped to improve the manuscript.
[1] Michaelian, K. Thermodynamics Stability of Ecosystems, J. Theo. Biol., 237 (2005) 323-335.
[2] Michaelian, K. (2016) Thermodynamic Dissipation Theory of the Origin and Evolution of Life: Salient characteristics of RNA and DNA and other fundamental molecules suggest an origin of life driven by UV-C light, Printed by CreateSpace, Mexico City, ISBN: 9781541317482, DOI: 10.13140/RG.2.1.3222.7443.
Reviewer 2 Report
You need to review your MS extensively, especially when it comes to English style.
Author Response
I thank the reviewer for reviewing my manuscript. I have made numerous corrections to the English style with the intention of improving the redaction.
Reviewer 3 Report
This paper is essentially identical to a paper recently submitted by the author to the journal Life. I do not detect any meaningful changes from the version that was submitted previously, therefore my review conclusions remain the same:
This paper presents a scenario for the origin of life based around an ocean surface environment, UVC radiation, fatty acid vesicles, and the synthesis of adenine from HCN and other precursors. The author discusses the absorption of UV radiation by the relevant molecules as an example of dissipative structuring, following the concept of dissipative structures introduced by Prigogine and coworkers. The author suggests that the UV absorption properties of many relevant biomolecules is not just related to self-protection, but to the effectiveness of dissipation of the UV energy to higher other energetic forms such as thermal energy.
The paper is well written and the photochemical properties of the relevant molecules are well discussed. The issue of UV light and the origins of life has been discussed at great length in the literature for many years (and the author cites several key publications in the field). Some in the field view UV light as inherently destructive to organic material and hence detrimental to any origins scenario. Others however, view the unique photochemical properties of biomolecules as a reflection of their ancient roots and indicators that UV radiation played a pivotal role in life's emergence. This debate will no doubt continue for many years to come, and contemporary knowledge and evidence are insufficient to conclude where the truth lies. This paper argues for a 'constructive' role of UV light in early life, specifically that the 'dissipative structuring' of UV light by HCN and other precursor molecules provided a photochemical energetic coupling mechanism, enabling several key endergonic reactions to occur, while destructive effects (various forms of molecular lysis) were sufficiently weak that synthesized molecules were not destroyed faster than they were formed.
There is already a large body of work investigating the interplay between the formation of HCN, UV light, and the synthesis of biomolecules, nucleobases in particular, but also sugars, fatty acids, and others. These studies include extensive experimental characterization of viable pathways and conditions. In the present paper, a simple numerical method is used to compute concentration profiles for the relevant molecules in a set of 24 reactions, leading to the production of adenine. This occurs in the context of fatty acid vesicles, assumed to be present in the ocean surface environment.
In my opinion, this paper is not suitable for publication for several reasons. Firstly, it does not provide any truly novel discoveries in terms of prebiotic synthesis (the results are from a simplified numerical model with strong assumptions, given the complex environment envisaged in the paper) or theoretical approaches to the origins problem. With regards to the numerical results, a series of reactions are modeled, but the many side reactions which would also occur in such a system are not mentioned or simulated. Much of the recent work in prebiotic chemistry has elucidated pathways for nucleobase synthesis from HCN and UV light. It is not clear how the present work achieves anything further than what has already been demonstrated in the literature, especially given that those works include experimental demonstrations. Furthermore, the title of the paper includes 'Evolution at the origin of life'. However, Adenine is just one nucelobase, its synthesis alone is not sufficient for abiogenesis, and there doesn't appear to be any notion of evolution in the presented results. The author discusses means for the synthesis of other nucleobases, but key phases still remain absent from this scenario, including the polymerization of nucleobases, sugars and phosphate to form a nucleic acid, the emergence of the ribosome and genetic code, etc. The author acknowledges on line 845 that the origins enigma will require more than just explanations for the synthesis of molecules. Most in the field would agree with this statement, but I do not see how the present paper presents more than a scenario for the synthesis of molecules (which in and of itself, is insufficient for the origin of life).
Secondly, the presented scenario includes fatty acid vesicles as a means of concentrating the relevant molecules (any ocean-based theory would suffer from a dilution catastrophe without some means of concentration). However, there was insufficient explanation as to the source of the fatty acids comprising such vesicles. There was a reference to high temperature hydrocarbon polymerization at hydrothermal vents, but this seems largely irrelevant to the present paper, because it is focused on an ocean surface environment, there would need to be an established delivery mechanism from one to the other. The second possibility that is referenced on line 422 is a self citation concerning possible mechanisms for fatty acid production by UV radiation. In my opinion, this is an insufficient basis for assuming the presence of fatty acid vesicles at the ocean surface.
The author discusses Prigogine's theories for dissipative structuring (which do not play a very significant role in modern thermodynamics, since they primarily apply to linear, near-equilibrium conditions, and do not include a description of information dynamics), but beyond the effective absorption of UV radiation, it is not clear how theories of dissipative structuring play any role in the scenario presented in the paper. Such theories are associated with pattern-forming systems such as convection cells and chemical oscillators. The author writes that the system presented is an example of nano-scale dissipative structuring, but still there is no evidence of emergent, organized patterning (even at the nanoscale) as would normally be associated with dissipative structures. Prigogone's work on non-equilibrium thermodynamics is normally associated with minimum entropy production in linear, close-to-equilibrium systems, and its validity in other conditions is still contested (some examples show that, e.g., Onsager's relations still work under certain far-from-equilibrium conditions). The author states on line 915 that the Prigogine theory does not constrain the total entropy production to increase or decrease. However a premise of the present paper is an apparent tendency to increase dissipation. Since dissipation is associated with free energy reduction and entropy production, it wasn't clear how there was still a connection to Prigogine's ideas (which the author stated do not constrain total entropy production). This needs clarification in the manuscript, is the Prigogine principle relevant or not, or is the fluctuation theorem more relevant. There should also be references and discussion of the fluctuation theorem derivations due to Crooks, Jarzynski and others. Concerning the selection of more dissipative phase space trajectories, there are potential connections to the work of Jeremy England and coworkers, and the probability theorist Edwin Jaynes.
Overall, despite being well presented, this paper does not appear to present any genuinely new findings that go beyond principles or results that are already in the contemporary literature.
Author Response
Response to Reviewer 3
I thank the reviewer for their revision of my manuscript and their suggestions for improvement. Indeed, an earlier version of my manuscript was sent to the Journal Life. I did not feel that it received a fair review in that journal since many of the criticisms were related to misunderstandings of the thermodynamic approach. Even though the editor afforded me the opportunity of re-submitting my manuscript after revision, the comments of a couple of the reviewers made me realize that the journal Entropy would have a more appropriate and appreciative audience.
The reviewer suggests that my manuscript is “well written and the photochemical properties of the relevant molecules are well discussed”. However, the referee suggests that “it does not provide any truly novel discoveries in terms of prebiotic synthesis (the results are from a simplified numerical model with strong assumptions,…”. I disagree with this analysis since, as I now make clear in the Introduction to the revised version, an important problem related to the origin of life is that of explaining within a consistent physical-chemical framework the dynamic origin and vitality of life, which includes not only molecular synthesis, but proliferation, maintenance, and evolution of fundamental molecules towards a complex biosphere. Proposals so far have been limited to supposing a kind of Darwinian evolution through selection based on molecular chemical stability or precursor sequestering ability, or assuming some fortuitous autocatalytic reaction set which took molecules to a sufficient level of complexity such that Darwinian natural selection would seem reasonable. These ideas have not led to a deeper understanding of the origin of life. I have suggested instead addressing the problem from non-equilibrium thermodynamic theory [1,2], in particular Classical Irreversible Thermodynamic framework of Prigogine and coworkers, in which the fundamental molecules arise naturally and spontaneously as microscopic dissipative structures (UV pigments) synthesized under the UV solar photon potential to dissipate this spectrum. Furthermore, I have shown how the concentration profile of those intermediate or product molecules would evolve over time to a profile which best dissipates the incident UVC spectrum.
What is new in the present manuscript, besides a detailed description of the photochemistry and non-equilibrium thermodynamics involved, is the application of this non-equilibrium thermodynamic formalism to the photochemical dissipative structuring of adenine from simpler and more common precursors. This has not been presented elsewhere before in any detail and I believe that it makes a valuable contribution to the field since it addresses the very important problem of explaining the dynamic vitality in the origin and evolution of life observable in the proliferation, metabolism of maintenance, and evolution of the intermediate and product molecular concentration profiles. This is now made clear in the first two paragraphs of the Introduction and Summary and Conclusions of the revised manuscript.
The reviewer expresses that concern that the notion of evolution is not explained in the text. The evolution is how the initial distribution of the concentrations of the precursor molecules evolves into the more complex and more dissipative final distributions of the concentrations of the final products which include adenine in significant quantities which thereby dissipates more effectively the imposed UVC solar photon spectrum. I have now emphasized this at the end of paragraph 4 of the section “Summary and Conclusions” in the revised version.
Fatty acid vesicles are presumed to have been in existence at, or very near, the origin of life by prominent researchers in the field [1]. There is, in fact, actual fossil evidence of such molecules dating from around 3.4-3.7 Ga [2] and reference to this data is now included in the revised version of my manuscript. As stated in the first paragraph of the subsection “Overview of the Model”, HCN and its hydrolysis product formamide are now recognized as probable precursors of fatty acids [3]. Furthermore, I have given reference to a published paper of ours which describes just how fatty acids may have been dissipatively structured within the same thermodynamic framework under the same UVC photon potential [4].
The reviewer questions the validity of applying Prigogine´s Classical Irreversible Thermodynamic formalism to the chemical/photochemical reaction model presented in my manuscript, suggesting that the theory has only limited application to near equilibrium linear regimes. This, in fact, is not true. Most of the examples in Prigogine’s numerous books are of chemical reactions which are inherently non-linear. Prigogine’s most recognized work was that of developing the formalism to treat the non-linear regime (see, for example, his 1967 book [5] and 1971 book with Glansdorff [6]). It is, in fact, only in the non-linear regime where dissipative structures arise. It was for this work in the non-linear regime that Prigogine was awarded the 1977 Nobel Prize in Chemistry, as can be ascertained from the Nobel committee’s specific remark , “… for his contributions to non-equilibrium thermodynamics, particularly the theory of dissipative structures.".
In the non-linear regime, multiple stationary states may exist (corresponding to particular distributions of the concentrations of the intermediate and final product molecules). In our model of differential equations for the chemical and photochemical reactions, a stationary state is observable when all concentrations of intermediate products reach fixed values in time (horizontal traces in figures 6-12). The system may evolve over stationary states, due to internal or external perturbations (provoked by passing the vesicle through regions of higher density of HCN in our model), since, unlike in the linear case, the stationary states in non-linear systems are only locally stable. The evolution over different stationary states is not deterministic, but rather probabilistic (induced by microscopic fluctuations near bifurcations), respecting always the Glansdorff-Prigogine criterion.
The entropy production may either increase or decrease during this evolution, but the overall tendency for non-linear systems with positive feedback (autocatalytic) systems is for the entropy production (or, specifically, photon dissipation) to increase. As explained in the manuscript, this has to do with the sizes of the catchment basin in a multidimensional phase space associated with each stationary state. The size of the catchment basins are related to access to conical intersections of the particular product molecule, which foment the dissipation of the photon energy into heat. (As an analogy from equilibrium molecular dynamics, the potential energy surface of a system may contain multiple minima. At a given fixed total energy the trajectory in phase space is such that the system spends most of the time around the catchment basin of the deepest minima since these generally have the largest catchment basins. )
There is, in fact, much evidence of dissipative structuring at the nanoscale dimension, for example in molecular machines and cellular organelles, and there is even a new field of research employing dissipative structuring to produce controlled nanoscale patterns for technological application [7]. In our model presented in the manuscript, the microscopic dissipative structures are the pigment intermediate and final products, and the corresponding macroscopic dissipative structures are the concentration profiles of these different intermediate and final products of the chemical/photochemical reactions. These distributions are not equilibrium distributions, they are instead distributions that are stable and maintained only under a constant UV solar flux and they tend towards increasing the dissipation of the incident photons.
The reviewer suggests that I reference the fluctuation theorem and other statistical mechanics treatments of evolutionary processes. In fact, I dedicated an Appendix to precisely this in the original manuscript. The original relevant analysis was not by any of the authors mentioned by the reviewer but rather by Onsager, Glansdorff, Prigogine, Nicolis and coworkers (the reviewer can verify this by studying, in particular chapter 8 of their 1971 book [6] and the paper [8], as well as the other references cited in the Appendix of my manuscript). As emphasized by Glansdorff and Prigogine, linear stability theory is only a simplified caricature of CIT formalism, corresponding to queries (fluctuations) only in the neighborhood of a stationary state and as such, contrary to what is occasionally claimed in the literature, linear stability theory is not sufficient to describe the evolution of non-linear dissipative systems.
I thank the reviewer once again for their comments and criticisms which have help to improve the manuscript.
[1] Oro, J. Chemical synthesis of lipids and the origin of life. J Biol Phys 20, 135–147, 1995. https://doi.org/10.1007/BF00700430
[2] Han, J. and Calvin, M. Occurrence of fatty acids and aliphatic hydrocarbons in a 3.4 billion-year-old sediment. Nature, 224 (5219) 576-577, 1969.
[3] Ruiz-Bermejo, M.; Zorzano, M.P.; Osuna-Esteban, S. Simple Organics and Biomonomers Identified in HCN Polymers: An Overview. Life, 3, 421–448, 2013. doi:10.3390/life3030421.
[4] Michaelian, K. and Rodríguez, O. Prebiotic fatty acid vesicles through photochemical dissipative structuring. Revista Cubana de Química, 31 (3), 354-370, 2019.
[5] Prigogine, I., 1967. An Introduction to the Thermodynamics of Irreversible Processes. Wiley, New York.
[6] Glansdorff, P., Prigogine, I., 1971 Thermodynamic Theory of Structure, Stability and Fluctuations, Wiley – Interscience.
[7] Soejima, T.,Amako,Y.,Ito,S.,Kimizuka,N. Light-reducible dissipative nanostructures formed at the solid-liquid interface. Langmuir, 30 (47) 14219–14225, 2014.
[8] Prigogine, I.; and Nicolis, G. Biological order, structure and instabilities. Quarterly Reviews of Biophysics 1247 (4), 107–144, 1971.
Reviewer 4 Report
The author has attempted to unify HCN chemistry, which may be driven by photochemical processes, and vesicle formation, which is required to support compartmentalization of reaction intermediates, in a Reaction-Diffusion model. The understanding of how structures such as purines can be synthesized under prebiotic conditions is certainly one of the wholly grails in the Origin of Life, and a thorough understanding on kinetic grounds has not yet been established. Despite his attempts, I believe that de author did not provide sufficient convincing evidences, as well as arguments, to claim “the first non-equilibrium thermodynamic analysis of the dissipative photochemical structuring of adenine (sentence 15)”.
My comments center around the specificity of various claims (minor revision) and validity of the analyses and conclusions of the manuscript (major revision).
Validity.
- On the mathematical model. The discussion (sentence 740) starts with the notion that table 4, and figures 6 and 7, ‘clearly’ indicates the formation of new stationary states. The referred table and figures in the current state, unfortunately, are not ‘clear’ as they are compiled with data that may or may not support this claim. I would appreciate it if the author could take the reader through what is visualized in these graphs, as it may be important for the story.
- I appreciate attempts to clarify the elementary reactions and the kinetic parameters in the list of assumptions on pages 15-19 but this list, unfortunately, does not validate the model described on page 20.
- The pH dependence appears to be omitted from the set of differential equations. For example, the hydrolysis of HCN to Formamide (reaction #1, and subsequently formic acid) occurs at strongly acidic solutions, whereas reactions 5-8 do not (see reference 51). Does reaction #1 occurs as suggested (at pH =7)? This seems to contradict to what is reported in reference 51.
- The assumption that the initial conditions chosen in Figure 6-12 would result in the dynamics as presented in this manuscript remains elusive. Truthfully, rate constants and conditions in references 51, 81, 82, are carefully chosen, and may not be amenable to the treatment as suggested in this work. Is it reasonable to have this network of reactions—without any side reactions?—to occur within a vesicle?
- I would suggest to, at least, use smaller parts of the model to simulate the values that are retrieved from these articles in an attempt to validate the accuracy of the model (in the absence of a vesicle). Do the concentration levels (micromolar concentrations) fit the experimental values that are obtained in the literature from which kinetic parameters were derived? Do the time-scales correlate with the reported synthetic procedures? Under which pH range would your model approximate the real solutions? How can you (or others) ensure that the vesicle remains in tact within the overall time span of the suggested syntheses?
- The author states that he has shown how 10-5 M adenine can be synthesized from 5 molecules (i.e., 10-22 M) of HCN, using a simple vesicle model and a system of photochemical reactions (sentence 804)”. I believe that such a claim is unfunded because:
- a time plot based on an unvalidated model does not provide prove of any synthesis may have happened. It has, thus, not been ‘shown’ nor ‘demonstrated’ that adenine can be synthesized from HCN. It may suggest that adenine can be produced from..
- It is unclear how can adenine be produced from a source molecule, HCN, with a concentration so close to zero (when taken mass-action kinetics in consideration).
- The author must justify why his manuscript enables him to draw conclusions 1 and 2. Which results support these claims remains elusive throughout the manuscript.
- Conclusion 3 assumes that micromolar concentrations of adenine is essential to life, or its origin.
- I would like to see literature references that supports this claim.
- Notwithstanding, even when literature supports this claim, the claim in the subsequent sentence (“such a scenario also establishes the foundation for the beginning of cellular life based on photon dissipation” 857) cannot be made based on the ‘observations’ made in this work.
- Finally, I find it hard to understand when assumptions are reasonable and when they are not (sentence 862). The author may elucidate the reasons or to tone down, or remove, claims that are not apparent from his work executed in this manuscript.
Specificity.
- Statements such as “it is generally believed (sentence 20)”, “numerous empirical evidences support (sentence 63)”and “there exist many proposal (sentence 114)” that have no references and points to a direction of generality that is unclear or may not exist. The author should consider adding references to these statements for clarification, and consider using different terms for ‘numerous’ and ‘many’.
- Furthermore, do to the lack of specificity, I find it difficult to understand some of the claims:
- “There are only two classes of structures in nature; equilibrium structures and non-equilibrium structures. (sentence 28)”. Perhaps, the author could mention a few examples—with references—of what is meant by these structures. Considering that convection cells is the only mentioned example, does the author perhaps meant states instead of structure?
- “Origin of Life was necessarily associated with ultraviolet photon absorption (sentence 163).” This conclusion does not follow from the preceding text in the same paragraph, and is difficult to support without a single reference to literature.
- “These photochemical and thermal reactions … will be considered in a future article (sentence 781)”. It is great to read that follow-up papers are in preparation but, to the point of specificity, I believe that the author should omit this sentence.
- ‘Fundamental’ molecules (in sentence 853 and 859). What does a fundamental molecule mean? Typo?
Overall, my central question is “to what extend is it reasonable to expect that combining kinetic parameters (that are retrieved under different conditions) could lead to a reaction-diffusion model as suggested in this manuscript?”. I have highlighted a series of examples of missing references and unsupported claims. There may be more occasions in this manuscript for which my comments of specificity and validity may apply.
With above, I am unable to support the publication of this manuscript.
Author Response
Response to Reviewer 4
I thank the reviewer for their honest analysis and useful criticism of my manuscript.
Here follow my responses to the reviewer’s comments under their heading of “Validity”. (I use the same numbering system employed by the reviewer.)
- Figures 6 and 7 represent the concentrations of the chemical constituents listed in table 1 as a function of time, as determined by the set of differential equations (Eqs.(9) to (20)) which represent the chemical and photochemical reactions occurring among the constituents under the imposed photon flux. New stationary states (horizontal traces at different, usually higher, concentrations) are reached after perturbations which correspond to the vesicle passing through regions of higher HCN concentration (0.01 M for 2 minutes every 3.5 days, evidenced in Fig. 6 by the vertical spikes). I have rewritten the first paragraphs of both the Results and the Discussion section of the revised version and also the caption of Fig. 6 to improve the understanding of the graphs.
2.
- As mentioned in the caption of Table 2, all reaction kinetic parameters were obtained at pH 7.0 except where no data were available from the literature, in which case I took the closest value of pH to neutral for which data was found, and then listed this particular pH value next to the reaction in Table 2. The kinetic equation for the hydrolysis of HCN to formamide (reaction #1), indeed was obtained at pH 7.0, as mentioned in the description of the reaction immediately after Table 2. These kinetic parameters were obtained from Kua and Thrush [82] using the experimental data of Miyakawa et al. [81]. A photon-induced tautomerization converts formamide (F) into formimidic acid (Fa). Basch et al. [72] (reaction #2).
Note that Sanchez et al. [51] preferred to do their experiments at pH range 8 to 9 (alkaline) because this gave them a greater ratio of polymerization of HCN over loss to hydrolysis, but this, according to the results of my reaction scheme at pH 7.0, is not necessary in order to obtain significant amounts of adenine. In fact, some amount of formimidic acid obtained from hydrolysis is useful as a catalyst for the production of amidine (Am) from formamide (F) (reaction #14), which supplies another route to adenine (reaction #15).
- Side reactions with HCN may occur, for example the formation of some of the amino acids (glycine, alanine, aspartic acid, serine) arise through hydrolysis of HCN [51] (something we plan to include in a future model). Other side reactions could also have been possible depending on what other molecules would have been available at the ocean surface. However, from many studies of HCN in pure water, the side reactions of most concern, because of their competition with HCN polymerization, appears to be hydrolysis, which we do take into account. Clearly, more experiments in more diverse environments would be important. However, many larger molecules would not be able to traverse the vesicle membrane and the affinities for diffusion transport of the smaller molecules over the membrane would depend on their concentrations within the vesicle and the environment. Any parasitic reaction within the vesicle consuming HCN would therefore lead to a compensating greater affinity for transport of this component across the membrane. A further point is that molecules with lower ionization energy or generally less strongly bound would be disassociated by the prevailing UVC light, both within the vesicle and within the external environment.
- Partial results of our model compare favorably with experimental data from the literature. These include the rate of hydrolysis of HCN compared to the rate of its polymerization to DAMN, which, according to the experiments of Sanchez et al. [51] are similar for concentrations of HCN (H) between approximately 0.01 M and 0.1 M (equal rates at 0.03 M for pH 7, T=80 °C). Also, the ratio of trans-DAMN (T) compared to cis-DAMN (C) under the given UVC flux (without permitting trans-DAMN (T) to be converted to AIAC (J)) arrives at a photostationary state with a value close to the experimental value of 4 determined by KochRodehorst [76]. Another partial result which follows experiment is that of the catalytic effect of trans-DAMN on the tetramization of HCN (reaction #7). In this case, the height in energy of the activation barrier was adjusted to give the catalytic effect observed for trans-DAMN in the experiment of Sanchez et al. [51]. In other cases, activation barrier energies and pre-exponential frequency factors employed were taken directly from experiment or fitting to experimental rate versus temperature graphs, or taken from accurate first principles calculations.
Published experiments leading to adenine from HCN in water under UVC light are few, piecemeal, and carried out under conditions considerably different from those assumed in the manuscript (e.g. most data are for high eutectic initial concentrations of HCN at low temperatures, alkaline pH, low pressure mercury lamp line emission at 253.7 nm) which make direct comparison with our adenine yields difficult (except for partial results as given above). However, we can compare our results with the quasi-equilibrium experiments of Ferris et al. [78]. Starting with a high concentration of HCN (0.1 M) in water (pH 9.2), and allowing this solution to oligomerize at room temperature for a 7 month period, and then subjecting these oligomers to hydrolyses at 110 °C for 24 hours, Ferris et al. obtain an adenine yield of 1 mg/l (equivalent to a concentration of 7.4e-6 M - the molar mass of adenine is 135.13 g/mol). Our model gives a similar adenine concentration of 2.6e-6 M but starting from a much lower concentration of HCN of only 6.0e-5 M (with only two perturbations of 0.01 M for two minutes each) and a more neutral pH of 7.0 within only 10 days, but at 90 °C and under a UVC flux integrated from 210 – 280 nm of about 4 W m^{-2} during daylight hours (figure 8). As another comparison, starting with an order of magnitude larger initial HCN concentration of 6.0e-4 M (and only one perturbation at 0.01 M for two minutes) at 80 °C gives us an adenine concentration of 7.6e-6 M after only 5 days (figure 10).
These results emphasize that our non-equilibrium reactor system with dissipative structuring under UVC light could lead to an important increase in adenine product compared to the equilibrium experiments performed without UVC light. As long as UVC light remains incident over a very dilute aqueous solution of HCN, then significant dissipative structuring of adenine could occur. There is no need to begin with large initial concentration of HCN by assuming low temperature eutectic conditions and there is no need for alkaline conditions in order to favor HCN polymerization over hydrolysis.
The above three paragraphs (under point 3) are included in the Discussion section of the revised manuscript, beginning on line 776.
The stability of lipid bilayers depends on many factors like chain length, amount of hydrogen saturation (conjugation), temperature, pH, and salt concentration. However, it is known that the stability of vesicles can be greatly increased through UVC-induced covalent cross linking between individual chains [1] which would be expected to occur naturally in our scenario. Such covalent cross linking at sites of conjugation on neighboring chains improves dramatically the stability over pH values, temperature, and salt concentrations [1]. Fatty acid vesicles are presumed to have been in existence at, or very near, the origin of life by prominent researchers in the field [2] and are often employed in models for the origin of life.
- The intended meaning of line 805 (old version of manuscript) was that one molecule of adenine could be synthesized from 5 molecules of HCN. I have changed the redaction of the Summary and Conclusions section to make it more understandable and concise and have removed references to “show” or “demonstrate” and instead use the word “suggest” as suggested by the reviewer
- -6. I have taken the reviewers suggestion to tone down and remove claims that are not apparent from the results. The summary of points in the Conclusions section of the old manuscript has now been replaced with shortened paragraphs in a revised section called “Summary and Conclusions”, which avoids including unsubstantiated claims.
Concentrations of up to 10^{-5} M of adenine were obtained with the reaction system considered in the manuscript after 30 days starting from the initial conditions specified in the caption of Fig. 6. Apart from synthesis of the nucleobases, other steps such as the formation of nucleosides, their phosphorylation and their condensation into oligos, replication of oligos, etc. would all have to be somehow related to increased photon dissipation if we are to claim that this suggested dissipative process leading to significant quantities of adenine could be relevant to the origin of life. We have published results of experiments, simulations, and deductions in this regard [3-6] and it is work still in progress. However, as mentioned above, we have now taken the reviewers suggestion and toned down the conclusions in a completely rewritten in the section called “Summary and Conclusions”.
The occurrence of all these processes, of course, would be favored if large concentrations of the nucleobases could be maintained against competing disassociation through hydrolysis or diffusion into the environment.
Response to the reviewers points of Specificity
- I have amended or removed all offending statements noted by the reviewer, or I have included references where they were needed.
2.
- I have improved the Introduction, particularly the description of the non-equilibrium dissipative structures (which can be defined as organization of material in space and time which occurs to increase the rate of the dissipation of an externally imposed thermodynamic potential) and give another example of both equilibrium structures and non-equilibrium structures in the revised version.
- The offending sentence at line 163 has been replaced with one more aligned with the hypothesis being tested in the manuscript.
- The sentence mentioning follow-up papers has been removed from the new version.
- The term “fundamental molecule” refers to those molecules which are found in all three domains of life; bacteria, archea, eukaryote. This is now defined at the first mention of this term in the Introduction.
The kinetic parameters for the reaction-diffusion model used in this manuscript have been chosen from the experimental and DFT simulation data available in the literature. Although their true values may differ somewhat from the values used, the general conclusions of this work, that the coupling of photochemical reactions with chemical reactions and diffusion under the non-equilibrium conditions brought about by the imposed UVC photon potential, leading to the auto-organization of dissipative structures, microscopic (pigments in the UVC) and macroscopic (pigment concentration profiles) which increase the dissipation of the imposed solar photon potential, would remain valid, independent of the uncertainties in parameter values. I have now included a statement to this effect in the Summary and Conclusions (line 853) of the revised version.
I once again thank the reviewer for their comments and criticisms which have helped to improve the manuscript.
[1] Y. Fan, Y. Fang, Ma L., and H. Jiang. Investigation of micellization and vesiculation of conjugated linoleic acid by means of self-assembling and self-crosslinking. J. Surfact. Deterg., 18:179{188, 2015.
[2] Oro, J. Chemical synthesis of lipids and the origin of life. J Biol Phys 20, 135–147, 1995. https://doi.org/10.1007/BF00700430
[3] Michaelian, K., Thermodynamic dissipation theory for the origin of life, Earth Syst. Dynam., 2 (2011) 37-51, doi:10.5194/esd-2-37-2011.
[4] Michaelian, K., Homochirality through Photon-Induced Denaturing of RNA/DNA at the Origin of Life, Life 2018, 8, 21.
[5] Michaelian, K. and Santillán Padilla, N. UVC photon-induced denaturing of DNA: A possible dissipative route to Archean enzyme-less replication, Heliyon 5, e01902 (2019). https://www.heliyon.com/article/e01902
[6] Mejia, J. and Michaelian, K., Photon Dissipation as the Origin of Information Encoding in RNA and DNA. Entropy 2020, 22, 940.
[51] Sanchez, R.A.; Ferris, J.P.; Orgel, L.E. Studies in Prebiodc Synthesis II: Synthesis of Purine Precursors and Amino Acids from Aqueous Hydrogen Cyanide. J. Mol. Biol. 1967, 80, 223–253.
[76] Koch, T.; Rodehorst, R. Quantitative investigation of the photochemical conversion of diaminomaleonitrile to diaminofumaronitrile and 4-amino-5-cyanoimidazole. J. Am. Chem. Soc. 1974, 96, 6707–6710.
[78] Ferris, J.; Joshi, P.; Edelson, E.; Lawless, J. HCN: a plausible source of purines, pyrimidines and amino acids on the primitive Earth. Journal of molecular evolution 1978, 11, 293–311.
Round 2
Reviewer 1 Report
The MS modification are beneficial. The importance of dissipative processes in the origin of life is now made much clearer and in my view this approach represents one of the major steps forward in addressing the OOL problem in recent years. I support this version for publication.
Reviewer 3 Report
The author has revised the manuscript based on comments from reviewers. Overall I do not feel the revisions are sufficient to justify the manuscript being published for several reasons. Firstly, the changes are not significant, they largely consist of additional sections of text that are similar to existing parts of the original manuscript. Secondly, the general thrust of the paper, that effective absorption of UVC light by HCN and compounds leading to adenine are somehow an explanation of the origin and evolution of life, are completely unsupported by the material in the paper. Thirdly, the presentation of contrived numerical models that depend on a large number of under-constrained parameters and are only loosely validated against experiment or other models, should not be used as evidence in support of a particular origins theory. Below are additional comments on the present manuscript, in addition to inline comments in the attached pdf:
The paper is divided into three sections: the thermodynamics of Prigogine and colleagues (which was highly controversial and does not enjoy considerable use by modern practitioners in the field), the photochemistry of relevant biomolecules, and a somewhat contrived (does not include the many side reactions that would have been present in the discussed scenario but perhaps don't play a role in the pathway that the author is concerned with) numerical model that conveniently demonstrates the author’s conjectures. The three sections of the paper are not particularly relevant to one another (e.g., the numerical model could be presented without the other two sections), largely independent, and only loosely connected by the text. I do not see a strong link between Prigogine's thermodynamics of minimum entropy production and the photochemical synthesis of adenine, since there is no evidence of the free forces adjusting such that the relevant contributions to entropy production are minimized (this is stated in lines 911 and 912 but not supported by a figure or result). In fact the total entropy production of the model system increases over time (as stated in the text), which cannot be connected to the principle of minimum entropy production, since that only applies to one term in the entropy equation (not the total entropy production)
Hydrothermal vent theories are misrepresented in this paper. They are not simply theories of ‘thermal’ reactions, they are, at the very least, theories of rock-water interactions, electrochemical reactions, redox reactions, pH gradients and mineral catalysis.
There is a significant assumption of fatty acid vesicles in this paper’s model system. This is justified by broad claims that 'fatty acids feature prominently in many origins scenarios'. While this is true, it is not sufficient justification in the present context and scenario. At the very least, a plausible chemical pathway should be suggested. In his response, the author wrote "HCN and its hydrolysis product formamide are now recognized as probable precursors of fatty acids [3]". However, this reference 3 due to Ruiz-Bermejo discusses formamide but makes no reference to fatty acids.
The paper assumes, without justification, that there are “patches of relatively high concentration (0.01 M) of HCN and formimidic acid”. Such a significant assumption must be appropriately justified or explained in the paper.
Another assumption that requires more detailed justification: “sufficient to permit a subsequent UVC polymerization of the purines into oligos assuming a possible UVC-assisted synthesis of ribose from similar precursor molecules [43] and a temperature [67] or formamide-catalyzed [68] phosphorylation“.
The legend lines of figure 6 are very difficult to decipher.
Figure 12 is overloaded with lines and colors and is very difficult to decipher.
The legend of Figure 15 needs to be re-positioned.
Table 4: the chemical labels on the top row need to be written out.
Given the recent progress in prebiotic chemistry, demonstration of adenine synthesis is no longer a particularly significant or novel result. Despite being a nucleobase, synthesis of this molecule alone does not imply explicit progress towards abiogenesis, since life is immeasurably richer and more complex than just a nucleobase in a vesicle. Given that experiments and models have already produced richer and more encouraging results, I see no persuasive reason that this manuscript would enhance the field in this respect.
The paper presents a set of reactions leading from HCN to adenine that are promoted by UVC light, and extrapolates from here to the idea that the complexity and evolution of the biosphere can be explained as extensions of this effect. The material presented in the paper are absoultely insufficient to support this very strong conclusion. Synthesis of a single biomolecule from precursors is a small step in a very long and complex journey from non-life to life. Extrapolating from this one step to a large swathe of the dynamics of life is simply unrealistic.
The author states in his response: “I believe that it makes a valuable contribution to the field since it addresses the very important problem of explaining the dynamic vitality in the origin and evolution of life observable in the proliferation, metabolism of maintenance, and evolution of the intermediate and product molecular concentration profiles.” What do you mean by “dynamic vitality”? Your paper simply ‘shows’ that adenine can be synthesised from HCN under a questionable set of initial and boundary conditions. This does not explain life’s ‘dynamic vitality’, and especially serves no role in explaining the evolution of early life, since it does not make significant progress towards the first genetic code, there is no mechanism for heritable variation among reproducing agents, and there is no integrated system of metabolism, compartment and information system. The author claims that because the product concentrations ‘evolve’ towards a distribution that more effectively absorbs the incident UVC light, that this constitutes a primitive form of evolution and is “more complex” than the initial distribution. But was the complexity of the initial and final distributions computed and presented? There is no quantitative statement of the complexity of the product distributions in the text and hence we have no way of knowing how the complexity of the product distributions changed over time. W.r.t. ‘evolving’ towards a distribution that better dissipates the incident UVC flux, firstly, this is not relevant to evolution in the biological sense, since there is no heritable variation, secondly, it is not surprising that such a distribution emerges since this dissipative capacity confers stability on those molecules that effectively dissipate the incident energy. These kinds of effects have been explored by Jeremy England and colleagues, whom were surprisingly not cited in the present work.
Overall, I do not support the publication of this paper. It largely repeats ideas that the author has already published, and does not present compelling evidence to support its conclusions, which in my opinion are a drastic oversimplification of the rich complexity of the biosphere.

Reviewer 4 Report
I am unable to support the publication of this manuscript. I believe that the author has made important changes to the main text but did not make significant changes in the results section nor answered most of my questions in the Referees Comments on previous manuscript (see attached below).
For example;
- The author made minor changes in the main text and Figures 6 and 7, whereas the question was to provide more details. The problem I have addressed in my previous comments were related to the lack of explanation of why the time traces are shaped as presented in Fig. 6 and Fig. 7. These should be one of most important figures as they underscore the statement below.
“These results emphasize that our non-equilibrium reactor system with dissipative structuring under UVC light could lead to an important increase in adenine product compared to the equilibrium experiments performed without UVC light. As long as UVC light remains incident over a very dilute aqueous solution of HCN, then significant dissipative structuring of adenine could occur. There is no need to begin with large initial concentration of HCN by assuming low temperature eutectic conditions and there is no need for alkaline conditions in order to favor HCN polymerization over hydrolysis.”
There is no apparent reason (given) for have oscillatory behavior in one of the components (Fig. 6) but none of the other. Could the author try to incorporate a reason? Any mechanisms to support this reason? Similarly, Fig. 7. Shows sustained oscillations in a few components but none of the others. Contrary to Fig. 6 though, these oscillations do not have consequences (as stated by the author).
The author also wrote:
“as can be seen by comparing figure 6 with 7 which was obtained without perturbation”
Does the reader need to see all the time traces or would a few suffice? Wouldn’t it be useful to show what the true/most important differences are between Fig. 6 and 7, and report on that, rather than ask the reader to compare two very congested graphs?
- The author did compare his model to literature value, but did not validate his model. I do appreciate the inclusion of the following comparisons and understand from the text that the author made attempts to validate his model.
“Partial results of our model compare favorably with experimental data from the literature. These include the… …gives us an adenine concentration of 7.6e-6 M after only 5 days (figure 10)”.
Having said so, I found it difficult to arrive at the same conclusions as the author did. To start, it would make it easier, and clearer, if a comparison is supported with figures rather than writing paragraphs that claim “partial results of our model compare favorably”. In here, the verbs ‘partial’ and ‘favorably’ have no specific meaning. Second, please see my comments below in [Validation] 2b, and 2c: I don’t think comparing to literature values alone explains these discrepancies.
- The author did add references but his introduction would benefit from having mentioned important players in the Origin of Life community and if proves come from research groups other than his own. As requested, the author should consider adding references to the highlighted statements (in previous comments) for clarification: Reference 3 is not sufficient as none of the authors such as Sutherland, Powner, Pross, Szostak, etc., etc., are mentioned. An option to include most of them would be to cite a book edited by A. Brack.. Reference 4-6 are self-citations and there are certainly more authors to be found at large that reports on various sources of free energy.
I could go on and check my own comments point-by-point but I believe that these three examples are sufficient to conclude that I remain unable to support the publication of this manuscript.
Referees Comments on previous manuscript.
The author has attempted to unify HCN chemistry, which may be driven by photochemical processes, and vesicle formation, which is required to support compartmentalization of reaction intermediates, in a Reaction-Diffusion model. The understanding of how structures such as purines can be synthesized under prebiotic conditions is certainly one of the wholly grails in the Origin of Life, and a thorough understanding on kinetic grounds has not yet been established. Despite his attempts, I believe that de author did not provide sufficient convincing evidences, as well as arguments, to claim “the first non-equilibrium thermodynamic analysis of the dissipative photochemical structuring of adenine (sentence 15)”.
My comments center around the specificity of various claims (minor revision) and validity of the analyses and conclusions of the manuscript (major revision).
Validity.
- On the mathematical model. The discussion (sentence 740) starts with the notion that table 4, and figures 6 and 7, ‘clearly’ indicates the formation of new stationary states. The referred table and figures in the current state, unfortunately, are not ‘clear’ as they are compiled with data that may or may not support this claim. I would appreciate it if the author could take the reader through what is visualized in these graphs, as it may be important for the story.
- I appreciate attempts to clarify the elementary reactions and the kinetic parameters in the list of assumptions on pages 15-19 but this list, unfortunately, does not validate the model described on page 20.
- The pH dependence appears to be omitted from the set of differential equations. For example, the hydrolysis of HCN to Formamide (reaction #1, and subsequently formic acid) occurs at strongly acidic solutions, whereas reactions 5-8 do not (see reference 51). Does reaction #1 occurs as suggested (at pH =7)? This seems to contradict to what is reported in reference 51.
- The assumption that the initial conditions chosen in Figure 6-12 would result in the dynamics as presented in this manuscript remains elusive. Truthfully, rate constants and conditions in references 51, 81, 82, are carefully chosen, and may not be amenable to the treatment as suggested in this work. Is it reasonable to have this network of reactions—without any side reactions?—to occur within a vesicle?
- I would suggest to, at least, use smaller parts of the model to simulate the values that are retrieved from these articles in an attempt to validate the accuracy of the model (in the absence of a vesicle). Do the concentration levels (mM concentrations) fit the experimental values that are obtained in the literature from which kinetic parameters were derived? Do the time-scales correlate with the reported synthetic procedures? Under which pH range would your model approximate the real solutions? How can you/we ensure that the vesicle remains in tact within the overall time span of the suggested syntheses?
- The author states that he has shown how 10-5 M adenine can be synthesized from 5 molecules (i.e., 10-22 M) of HCN, using a simple vesicle model and a system of photochemical reactions (sentence 804)”. I believe that such a claim is unfunded because:
- a time plot based on an unvalidated model does not provide prove of any synthesis may have happened. It has, thus, not be ‘shown’ nor ‘demonstrated’ that adenine can be synthesized from HCN. It may suggest that adenine can be produced from..
- It is unclear how can adenine be produced from a source molecule, HCN, with a concentration so close to zero (when taken mass-action kinetics in consideration).
- The author must justify why his manuscript enables him to draw conclusions 1 and 2. Which results support these claims remains elusive throughout the manuscript.
- Conclusion 3 assumes that micromolar concentrations of adenine is essential to life, or its origin.
- I would like to see literature references that supports this claim.
- Notwithstanding, even when literature supports this claim, the claim in the subsequent sentence (“such a scenario also establishes the foundation for the beginning of cellular life based on photon dissipation” 857) cannot be made based on the ‘observations’ made in this work.
- Finally, I find it hard to understand when assumptions are reasonable and when they are not (sentence 862). The author may elucidate the reasons or to tone down, or remove, claims that are not apparent from his work executed in this manuscript.
Specificity.
- Statements such as “it is generally believed (sentence 20)”, “numerous empirical evidences support (sentence 63)”and “there exist many proposal (sentence 114)” that have no references and points to a direction of generality that is unclear or may not exist. The author should consider adding references to these statements for clarification, and consider using different terms for ‘numerous’ and ‘many’.
- Furthermore, do to the lack of specificity, I find it difficult to understand some of the claims:
- “There are only two classes of structures in nature; equilibrium structures and non-equilibrium structures. (sentence 28)”. Perhaps, the author could mention a few examples—with references—of what is meant by these structures. Considering that convection cells is the only mentioned example, does the author perhaps meant states instead of structure?
- “Origin of Life was necessarily associated with ultraviolet photon absorption (sentence 163).” This conclusion does not follow from the preceding text in the same paragraph, and is difficult to support without a single reference to literature.
- “These photochemical and thermal reactions … will be considered in a future article (sentence 781)”. It is great to read that follow-up papers are in preparation but, to the point of specificity, I believe that the author should omit this sentence.
- ‘Fundamental’ molecules (in sentence 853 and 859). What does a fundamental molecule mean? Typo?
Overall, my central question is “to what extend is it reasonable to expect that combining kinetic parameters (that are retrieved under different conditions) could lead to a reaction-diffusion model as suggested in this manuscript?”. I have highlighted a series of examples of missing references and unsupported claims. There may be more occasions in this manuscript for which my comments of specificity and validity may apply.
With above, I am unable to support the publication of this manuscript.